# Soilscape evolution of aeolian-dominated hillslopes during the Holocene: investigation of sediment transport mechanisms and climatic-anthropogenic drivers

Sagy Cohen[1,2*], Tal Svoray[2], Shai Sela[2], Greg Hancock[3] and Garry Willgoose[4]

[1] Department of Geography, University of Alabama, Box 870322, Tuscaloosa, Alabama 35487, USA.

[2] Department of Geography and Environmental Development, Ben-Gurion University of the Negev, Israel.

[3] School of Engineering, The University of Newcastle, Callaghan, New South Wales 2308, Australia

[4] School of Environmental and Life Sciences, The University of Newcastle, Callaghan, New South Wales 2308, Australia

* Corresponding author:

Email: sagy.cohen@ua.edu; Phone: 1-205-348-5860; Fax: 1-205-348-2278

Keywords: Soilscape, Pedogenesis, Sediment Transport, Modeling, Aeolian, Loess.

Submitted to: Earth Surface Dynamics

**Abstract**

Here we study the soilscape (soil-landscape) evolution of a field-site at the semiarid zone of Israel. This region, like similar regions around the world, was subject to intensive loess accumulation during the Pleistocene and early Holocene. Today, hillslopes in this region are dominated by exposed bedrock with deep loess depositions in the valleys and floodplains. The drivers and mechanism that led to this soilscape are unclear. Within this context, we use a soilscape evolution model (mARM5D) to study the potential mechanisms that led to this soilscape. We focus on advancing our conceptual understanding of the processes at the core of this soilscape evolution by studying the effects of fluvial and diffusive sediment transport mechanisms, and the potential effects of climatic and anthropogenic drivers. Our results show that in our field site, dominated by aeolian soil development, hillslope fluvial sediment transport e.g. surface wash and gullies, lead to downslope thinning in soil while diffusive transport e.g. soil creep lead to deeper and more localized soil features at the lower sections of the hillslopes. The results suggest that, in this semiarid, aeolian-dominated and soil depleted landscape, the top section of the hillslopes is dominated by diffusive transport and the bottom by fluvial transport. Temporal variability in environmental drivers had a considerable effect on soilscape evolution. Short but intensive changes during the late Holocene, imitating anthropogenic landuse alterations, rapidly changed the site's soil distribution. This leads us to assume that this region's soil depleted hillslopes are, at least in part, the result of anthropogenic drivers.

## 1. Introduction

Southern Israel, similar to other regions around the world, was subject to intensive loess accumulation during the Pleistocene and early Holocene. Hillslopes in this region are currently dominated by exposed bedrock with deep loess deposits in the valleys. The drivers and timing of the soilscape evolution that led to this soilscape are debatable. Studies in southern Europe and in the northern parts of the Middle East have found that anthropogenic activities (e.g. shrub removal, logging/timber extraction and over grazing in the late Holocene) were the dominant driver for the extensive removal of soils from hillslopes in many regions (*Fuchs et al*., 2004; *Fuchs*, 2007; *van Andel et al*., 1990). These conclusions differ from studies in the Negev Desert in Israel which found that most of the hillslope loess apron was eroded in the early Holocene, prior to significant human settlement (*Avni et al*. 2006). This finding suggests that the degradation of soil from the Negev Desert hillslopes, where such existed, was driven by climatic, rather than anthropogenic processes. Consequently, there is an ongoing debate in the literature regarding the drivers of the extensive soil depletion in Mediterranean and southern European hillslopes.

From a soilscape evolution point of view, aeolian dominated soilscapes differ from bedrock-weathering dominated soilscapes in several ways. In bedrock-weathering systems *in situ* weathering rates decrease exponentially with soil depth (*Gilbert*, 1877; *Ahnert*, 1977), thus regulating soil production as a function of regolith thickness (*Heimsath et al*., 1997). Weathering of regolith and soil leads to vertical particle size distribution with finer particles closer to the surface as a

function of the soil and regolith age, namely time exposed to weathering (*Yoo and Mudd* 2008). At the surface, armouring can develop by size-selective entrainment (*Kim and Ivanov*, 2014) or vegetation shielding, which limits sediment transport by overland flow (*Willgoose and Sharmeen*, 2006). Given sufficient time and in the absence of vertical mixing due to pedoturbation, these processes - depth dependent weathering, vertical self-organization and surface armouring - will stabilize the soilscape leading to steady-state or dynamic equilibrium conditions (*Cohen et al.*, 2013 & 2015). In aeolian dominated landscapes these controls on soil production and transport are largely ineffective as: (1) much of the soil is transported to the system as airborne sediments, i.e., no depth dependency; and (2) fine and highly erodible material is continuously deposited on top of older surface soils which limits the potential for surface armouring and vertical self-organization.

The differences between aeolian and bedrock-weathering dominated soilscapes lead us to conclude that traditional (i.e. bedrock weathering originated) soilscape evolution analysis is inappropriate for investigating the history of the aforementioned loess soilscapes. In *Cohen et al.* (2015) we developed a soilscape evolution model (mARM5D) to study the differences and interactions between aeolian and bedrock weathering soil production on a synthetic 1D hillslope. In that paper we have found that bedrock weathering dominated soilscapes are considerably more stable and showed much lower spatial (aerial) variability in soil depth and particle size distribution (PSD). We proposed that aeolian-dominated landscapes are more responsive to environmental changes (e.g., climatic and anthropogenic) compared with bedrock-weathering landscapes.

Here we use mARM5D to investigate an aeolian dominated field-site in central Israel located at the margin between Mediterranean and arid climates and with long history of human settlement. We introduce anthropogenic and climatic drivers to investigate the potential importance of temporal dynamics on soilscape evolution. We focus our analysis in this paper on the differences between Fluvial (rilling, hillslope wash and concentrated flow) and Diffusive (soil creep) hillslope sediment transport mechanisms. We seek to gain better understanding about how these sediment transport mechanisms affect soilscape evolution in this soilscape. This is important as: (1) each transport mechanism is affected differently by climatic/anthropogenic drivers; and (2) we do not know what is the potential contribution/importance of each mechanism on soilscape evolution.

## 2. Methodology

### 2.1 Field site and measured data

The field-site (Long Term Ecological Research, LTER, near Lehavim in the Northern Negev, Israel; $31^0 20'$ N, $34^0 45'$ E; Figure 1) is situated on the desert margin between a Mediterranean climatic regime to the north and an arid climatic regime to the south (note changes in green vegetation in Figure 1a). The area of the site is 0.115 km$^2$. This region has shifted between these two climatic regimes throughout the Pleistocene and Holocene (*Vaks et al.*, 2006). This region has also seen varying degrees of human settlement and agricultural activity throughout the late Holocene. The history of this region (both

human and natural) gives us a unique opportunity to study how climatic and anthropogenic drivers may have affected hillslope geomorphology resulting in the soil-depleted landscape we see today.

The LTER site is located in Aleket basin with an average rainfall of 290 mm per annum. The mean annual temperature is 20.5$^0$C, with a maximum of 27.5$^0$C and a minimum of 12.5$^0$C. The terrain is hilly and the area is divided by an east-west flowing ephemeral stream. The dominant rock formations are Eocenean limestone and chalk with patches of calcrete. Soils are brown lithosols and arid brown loess. Much of the loess was eroded from the hillslopes and deposited in the valleys (several meters deep in some locations). The vegetation is characterized by scattered dwarf shrubs (dominant species *Sarcopoterium spinosum*) and patches of herbaceous vegetation, mostly annuals, are spread between rocks and dwarf shrubs (*Svoray et al*. 2008). The herbaceous vegetation is highly diverse, mostly composed of annual species (*Svoray and Karnieli 2011*). At the research site a typical convex shaped slope was chosen for testing model predictions (Figure 1d).

A dataset of measured topography and soil parameters at the study site (including soil depth distribution and a Digital Elevation Model; DEM) is available from a previous study (*Sela et al*. 2012). A soil depth map (Figure 2) was compiled using Ordinary Kriging interpolation of 550-point measurements. An orthophoto (at 10 cm$^2$ pixel resolution) was use to classify exposed rock and assign zero depth to the interpolation map. The DEM used in this study was obtained from 700 measured points (at approximately 10 meter intervals) using a laser theodolite (SOKIA Inc. Total Station) and interpolated using Ordinary Kriging to a horizontal resolution to 2 x 2 meter pixel resolution for the mARM5D simulations. From this DEM a D8 flow direction, Dinf (D-infinity algorithm; *Tarboton,* 1997) slope (m/m) and Dinf contributing area layers were calculated using the TauDEM tool (Tarboton, 2010).

## 2.2 Application of mARM5D to Lehavim site

In *Cohen et al*. (2015) we developed a dynamic soil evolution model (mARM5D) to simulate soil physics as a state-space system as an extension of the mARM3D model (*Cohen et al*., 2009; 2010). mARM5D is a modular and computationally efficient modelling platform that explicitly simulates three spatial dimensions in addition to a temporal dimension and a PSD (hence the 5D suffix). The cellular model simulates soil evolution over a given landscape by describing changes in PSD in a finite number of equally thick soil profile layers (size and number are defined by the user) in each grid-cell.

The mARM framework introduced a novel implementation of physically-based equations using transition matrices that express the relative change in spatially and temporally explicit PSD vectors. This concept greatly improves the model computational efficiency and modularity but is challenging to describe in full. Below we describe the mARM5D physically-based equations that include the parameters that are modified in the simulation scenarios we analysed in this paper. A full description of the mARM model architecture as a platform to mARM5D can be found in the following publications: The model weathering component was explored in Cohen et al. (2010), its spatiotemporal algorithms in Cohen et al. (2013) and

its aeolian and sediment transport components in Cohen et al. (2015). In Cohen et al. (2015) also outline and discuss the model assumptions.

Here, we simulate the spatial and temporal changes in PSD as resulting from: (1) physical weathering of bedrock and soil particles in each profile-layer; (2) aeolian deposition on top of the surface layer; (3) size-selective entrainment and deposition by overland flow (generally referred to here as fluvial sediment transport) from/on the surface layer; and (4) non size-selective diffusive sediment transport (creep) both on the surface and within the soil profile.

### 2.1.1 Fluvial transport

For each grid-cell, the top layer is the surface layer exposed directly to size-selective erosion. Sediment transport capacity over a timestep ($q_s$, m$^3$/m) at the surface is calculated using a modification of the TOPOG model (TOPOG, 1997; Merritt et al., 2003) sediment transport equation

$$q_s = e \frac{q^{n_1} S^{n_2}}{(s-1)^2 d_{50}^{n_3}} \Delta t \qquad (1)$$

where $e$ is an empirical erodibility factor, $q$ is discharge per unit width (m$^3$/s/m), $S$ is slope (m/m), $d_{50}$ is the median diameter (m) of the material in the surface layer, $s$ is the specific gravity of sediment ($s$=2.65; kg/m$^3$), $n_1$, $n_2$ and $n_3$ are calibration parameters and $\Delta t$ is the iteration timestep size (e.g. 0.1 year). The units of erodibility parameter $e$ are a function of the calibration exponents n$^1$, n$^2$ and n$^3$ and are defined such that the units of $q_s$ are the ones specified. We used here $n_1$=1 and $n_2$=1.2 based on a calibration in *Cohen et al.* (2009) and modified $n_3$ to 0.5 (from 0.025) to adjust for the very fine-grained aeolian sediment.

Discharge ($q$; m$^3$/s/m) is

$$q = \left[ \frac{A}{A_p} \right]^{n_4} \frac{Q}{(A_p)^{0.5}} \qquad (2)$$

where $Q$ (m$^3$/s) is the excess hillslope runoff variable, $A$ is the upslope contributing area (m$^2$), $A_p$ is the area of a grid cell unit (m$^2$) and $n_4$ is a constant relating runoff as a function of contributing area. In *Cohen et al.* (2010 and 2015) the relationship between contributing area and runoff discharge was assumed to be linear ($n_4$=1). This assumption could not be justified in our field-site as *Yair and Kossovsky* (2002) showed that runoff generation in this region does not increase linearly downslope. Using an extensive parametric study (not presented here) we have found that $n_4$=0.1 leads to best approximation of observed soil distribution. We will discuss this later. Water is routed to a neighbouring grid-cell with the 'steepest descent' (D8) algorithm (*O'Callaghan and Mark*, 1984).

### 2.1.2 Diffusion transport

Traditionally, equations of two-dimensional diffusive transport calculate sediment discharge as a linear relationship to slope, soil thickness and a diffusion coefficient (e.g. the creep model of *Culling*, 1963, or the viscous flow model of *Ahnert*, 1976) and, if the soil is explicitly modelled at all, diffusion is considered independent of depth through the profile. Simulation of the soil profile in mARM5D is novel as it explicitly calculates diffusive transport for each soil profile layer. Based on *Roering* (2004), the diffusivity is assumed to decrease exponentially with depth below the soil surface:

$$Dc_l = \exp(-\lambda h_l) \tag{3}$$

where $Dc_l$ is the fraction of diffusion rate for the layer $l$ relative to the diffusion rate at the surface layer ($l_s$), $h_l$ is the mean depth (m) of profile layer $l$ relative to the surface and $\lambda$ is a calibration parameter. We used $\lambda$=0.02 based on *Fleming and Johnson* (1975) and *Roering* (2004). The surface diffusion sediment transport rate ($D_s$; m) is:

$$D_s = \left(\frac{s}{s_a}\right)^{\beta} D_o \Delta t \tag{4}$$

where $D_o$ is the surface diffusivity (m/y) and $S_a$ is the adjustment slope, the average slope in which $D_o$ was measured/estimated. Here we use $S_a$=0.2 which approximate our field site average slope. Using an extensive sensitivity analysis we have found that $\beta$=0.1 yielded the best approximation to our field site's soil distribution. This value differs from the typical assumption of a linear relationship between slope and diffusion ($\beta$=1), suggesting that the influence of topographic slope in this soilscape is much lower. We will discuss this later. The removal of material due to diffusion from a given layer is calculated as the proportion of the layer's movable material (expressed in the model as a PSD vector $\underline{g_l}$) that has been displaced at each iteration:

$$\underline{g_l}_{t+1} = \underline{g_l}_t - \left[\underline{g_l}_t \left(\frac{D_s}{\sqrt{A_p}}\right) Dc_l\right]. \tag{5}$$

### 2.1.3 Aeolian deposition

Sediment, with a user-defined grading distribution ($\underline{g}_a$), is added to the surface layer. The aeolian deposition rate ($K_a$; mm/yr) is assumed to be spatially uniform:

$$h_s \underline{g_s}_{t+1} = h_s \underline{g_s}_t + K_a \underline{g}_a \tag{6}$$

where $\underline{g_s}_t$ is the vector for the surface layer PSD and $h_s$ is the thickness of the surface layer. We use the same PSD as in the fine-grained simulation in Cohen et al., (2015), with a $d_{50}$=0.06 mm (derived from Bruins and Yaalon, 1992)

. For the sake of simplicity, aeolian sediment is assumed to originate from outside the system and no aeolian erosion is considered within the simulated domain. This means that $K_a$ is, in our case, the aeolian sediment accumulation (deposition) rate.

**2.3 Simulation scenarios**

Four model parameters are driven by climate and anthropogenic changes:

1. $e$ - Surface Erodibility (equation 1);

2. $Q$ – Runoff (equation 2);

3. $D_0$ – Surface diffusive transport rate (equation 4);

4. $K_a$ – Aeolian deposition rate (equation 6).

The effect of climate and anthropogenic change on the model parameters represent our best estimates based on the literature for this semiarid region. They can be summarized as:

1. Wetter climatic conditions allow for higher vegetation cover and thus lower surface erodibility and runoff generation ($e$ and $Q$ respectively) (*Goodfriend*, 1987, *Zilberman*, 1992 and *Avni et al.*, 2006).

2. During wetter climatic condition colluvial processes are more intensive (*Goodfriend*, 1987 and *Zilberman*, 1992), translating into a higher diffusive sediment transport rate ($D_0$).

3. During wetter climatic condition aeolian deposition rates are higher ($K_a$) (*Horowitz*, 1979 and *Bowman et al.*, 1986).

4. Human activities in this area reduce vegetation cover on the hillslopes (mostly by grazing), enhancing the effect of the dry climate during the Holocene (*Fuchs et al.*, 2004), increasing $e$ and $Q$ and decreasing $D_0$ and $K_a$.

Using these assumptions we divided the simulation scenario into three homogenous periods (Figure 3 and Table 1) based on *Vaks el al*. (2006):

P1- Late Pleistocene (80-12 kyr BP): wetter climatic period – a factor of 0.1 for erosivity and runoff ($e$ and $Q$ respectively), scale of 2 for diffusion ($D_0$) relative to modern rates and a maximum rate for aeolian deposition ($K_a$).

P2 - Early Holocene (12-8 kyr BP): dry climatic period - scale of 0.2 for $e$ and $Q$, factor of 2 for $D_0$ (unchanged from P1) relative to modern rates and scale of 0.5 for $K_a$ relative to its P1 (maximum) rate.

P3 - Late Holocene (8-0 kyr BP): increasingly drier climate with human activity - scale of 1 (maximum) for $e$ and $Q$, scale of 1 for $D_0$ and factor of 0.1 for $K_a$ relative to its P1 (maximum) rate.

**2.4 Simulated processes and calibration**

Three site-scale simulations are analyzed in this paper:

S1- sediment transport is simulated only by fluvial processes;

S2- sediment transport is simulated only by diffusion;

S3- sediment transport is simulated by both diffusive and fluvial mechanisms.

Soil is produced and supplied by both bedrock weathering and aeolian deposition. Soil production by bedrock weathering was assumed to be small relative to loess accumulation rate due to the dominance of limestone geology in the site. Limestone bedrock typically results in limited soil production by weathering except for producing a Mollisol, which is not simulated, and rock fragments, which are simulated. Weathering rate ($P_0$ in equation 5) is thus set to spatially and temporally constant value of 0.01 mm/y. Maximum aeolian deposition rate (during P1 simulation scenario period) is spatially constant and set to 0.1 mm/y based on *Bruins and Yaalon* (1992).

Initial values during the P3 period (most modern) for $Q$ was estimated based on *Eldridge et al*. (2002) and *Yair and Kossovski* (2002) and for $D_0$ based on *Carson and Kirkby* (1972). Adjusting these two parameters controls the ratio between the fluvial and diffusive sediment transport mechanisms. The values of these parameters were refined by an extensive parametric study to best match observed soil depth distribution. The best match was for $Q$=0.0066 m$^3$/y and $D_0$=6 mm/y for the P3 period (Table 1).

For the S1 and S2 simulations, the $Q$ and $D_0$ parameters were adjusted to yield a similar average soil depth as the S3 simulation. This adjustment ensures that the differences observed between the three simulations are mainly due to differences in sediment transport mechanism, not the accumulative variations in sediment transport rate. For S1 $Q$ was adjusted to 0.017 m$^3$/y and $D_0$ was set to 0 (no diffusive transport; Table 1). For the S2 simulation $D_0$ was adjusted to 10.75 mm/y and $Q$ was set to 0 (no fluvial transport).

## 3. Results

### 3.1 Field Site Application

For the fluvial simulation (S1), the P1 period, with low runoff and surface erodibility and high aeolian deposition (Figure 3), produced deep soils on the hillslopes (up to 200 cm; Figure 4a-b). During P2, with higher runoff and surface erodibility rates and lower aeolian deposition rate (by a factor of 2), soil is slowly eroding primarily from the lower sections of the hillslopes (Figure 4c-d). Erosion greatly intensifies during P3 due to further increase in runoff and surface erodibility rates and lowering in aeolian deposition rate (by a factor of 5). By the end of P3, most of the thick hillslopes loess apron has been eroded (Figure 4f) leaving two clusters of relatively deep soils (about 150 cm deep) on the interfluve, as well as quite extensive shallow aprons (about 50 cm deep) at the top and middle sections of the hillslopes. The rest of the hillslope is covered with a shallow soil layer (<20 cm) with no exposed bedrock. This soil distribution does not correspond well with observed soil depth (Figure 2) which exhibits a high degree of exposed bedrock at the top section of the hillslopes and the interfluve and deeper soils at the lower parts of the hillslopes.

The diffusive simulation (S2) yielded long straight bands of soil deposition along parts of the simulation domain (Figure 5). These bands follow the D8 flow direction input and are only apparent in the diffusive simulation. This is because soil transport away from a grid cell is not affected by its upstream contributing area (only its local slope) for the diffusive mechanism while deposition will is higher in cells with greater flow accumulation (more sediment has the potential of being transported to it). As a result cells along concentrated flow paths may result in deep soil deposited from upstream cells. S2 is an extreme diffusion scenario, combining highly mobile sediment influx (aeolian deposition) with high diffusion rates (enhanced by the high topographic slopes in this field site). While more moderate landscapes and rates will minimize these artefacts improvement to the diffusion transport mechanism is likely needed and be the focus of future research.

The P1 period for the S2 simulation, with high diffusive and aeolian deposition rates, produced soil accumulation at lower sections of the hillslopes (Figure 5a-b). These deposition features are over 100 cm deep at the footslope and are decreasing in depth upslope. The upper sections of the hillslopes are covered with a shallow loess apron (< 20 cm) with narrow bands of exposed bedrock (white color) along the interfluve. During P2, aeolian deposition rate decreased while the diffusive rate remained high (Figure 3). This leads to erosion of the upslope deposition bands resulting in a slight decrease in their spatial extent (Figure 5c-d). The extent of the exposed bedrock feature along the crest and down the hillslopes increased. During P3, the diffusive rate decreases by a factor of 2 and aeolian deposition by a further factor of 5. The main impact of this reduced soil supply is an extensive degradation of the thin loess apron on the hillslopes (Figures 5e-f). The deposition bands at the bottom of the hillslopes are relatively unaffected. Final soil distribution (Figure 5f) better corresponds with the measured soil distribution (Figure 2) compared with the S1 simulation (Figure 4f). Measured soil depth tends to be more heterogeneous and widespread and does not show extremely localized deposition features at the footslopes.

In the combined fluvial and diffusive simulation (S3) the P1 period shows deep soil features, about 150 cm, covering most of the intermediate and lower sections of the hillslopes (Figure 6a-b). With an exception of a thin band of deep soils near the crest, the upslope parts are covered with a shallow loess apron (less than 15 cm). The changes during P2 (aeolian deposition decrease by a factor of 2, fluvial rate increase by a factor of 2 and diffusive rate remain high) initially led to degradation of the loess apron at the upper parts of the hillslopes (Figure 6c). Once the loess apron has been completely removed, the deposition features at the lower section of the hillslopes start to erode (Figure 6d). This trend accelerates during P3 due to the sharp decrease in aeolian deposition rate. The increase in fluvial rate by a factor of 5 while diffusive rate decreases by a factor of 2 (Figure 3) leads to greater erosion at the bottom parts of the deposition features (Figure 7e-f). The resulting soil distribution better corresponds with measured soil distribution: exposed bedrock at the top and bottom parts of the hillslopes with a mostly shallow band of soil at the middle part of the hillslopes. The considerable changes in soil depths during P3 shows that intense but relatively short changes in external drivers (representing anthropogenic alterations in this study) can be significant for this soilscape evolution.

## 3.2 Transect (1D) Analysis

Soil depth evolution was plotted along a transect on the northwestern facing hillslope (thick black line with crossing short lines in Figures 1c and 4-6), focusing on the last 16 kyr of the simulations (the most dynamic period of these simulations).
The S1 profile (Figure 7a) gradually thins toward the footslope while the S2 (Figure 7b) profile is very thin at the top of the hillslope and then thickens considerably from nearly zero depth to about 190 cm over a stretch of less than 5 m. The S3 simulation (Figure 7c) has also resulted in a steep step in soil depth between the upper and lower parts of the hillslope. However, the S3 simulation resulted in considerable variability in soil depth at the footslope. In the S1 simulation the hillslope profile changes considerably during the plotted 16 kyr while the S2 profile displays only minor variation and S3 varies mainly at the bottom of the hillslope. The S1 hillslope profile initially erodes evenly in space but during P3 it shows increased erosion rate at the top of the hillslope. This trend is also visible in S3.
The final (0 kyr BP) hillslope profile for S3 has a nearly 35 m long exposed bedrock section at the top part of the hillslope (also visible in Figure 6f) followed by a deposition section with a downslope decreasing soil depth (from about 100 to 10 cm at the bottom of the hillslope). This profile has a number of both steep and shallow steps in soil depth (from more than a 100 cm to less than 10 cm) which are commonly observed in the Lehavim field-site. The measured soil depth profile along the transect (Figure 7d) is shallower and displays a smoother (it is an interpolation of measurement points) transition between the erosive and deposition parts of the hillslope. Overall the S3 soil-depth profile (Figure 8c) shows similar trends to the measured soil depth (Figure 7d). Particularly notable is the correspondence in the location of the mid-slope soil depth depression.

## 4. Discussion

*Roering* (2008) simulated soilscape evolution in a soil-mantled and vegetated landscape in northwestern U.S. He studied a number of diffusive sediment transport mechanisms and found that the best fit for the observed landform and soil distribution (increase in soil depth with slope angle downslope) was a nonlinear and soil depth-dependent model. He argued that soil thickness controls the magnitude of biogenetic activity (e.g. bioturbation) which drives sediment transport. In semiarid soil-depleted environments, landscape evolution and soil distribution was also found to be related to soil depth but by a different mechanism. *Saco et al.* (2007) simulated a semiarid and soil-depleted landscape and showed that soilscape evolution under water-limited conditions tend to follow a source-sink dynamics in which soil bands (which are deep enough to support vegetation) will act as a sink for water and sediment fluvially transported (surface wash) from bare intermediate sections between the vegetated bands.
The Lehavim LTER field site, under modern climatic conditions, is under a water-limited regime resulting in vegetation patches acting as sinks to the exposed bedrock section on the hillslope (*Svoray et al.*, 2008; *Svoray and Karnieli,* 2011) leading to micro-topographic variability. The site's soilscape is also characterized by a general trend of increasing soil depths

downslope. This suggests an intriguing interplay between semiarid and soil-mantled soilscape evolution. Our results show that during wetter periods (with greater aeolian deposition rates, P1) soil was thickening at the downslope direction (Figure 5b and Figure 6c). During the following drier periods (P2 and P3) the bottom part of the hillslopes started eroding resulting in a soil distribution where the middle part of the hillslope shows the deepest soil (the mid-slope depression; Figure 5 and 6).

Simulated fluvial sediment transport led to a somewhat unusual soil distribution in which soil is thickest at the interfluve (Figure 4). Hints of this kind of soil distribution are evident in the Lehavim LTER site (e.g. north sections of both northwest and southeast facing hillslopes; Figure 2) but are not as prominent as the S1 simulation predicted. Diffusive transport tended to produce localized deep deposition features at the bottom of the hillslopes. Evidence of this type of soil distribution can also be seen in this field-site (primarily on the southeast facing hillslope; Figure 2) though not as deep and localized as the

S2 simulation (diffusive only sediment transport; Figure 5) produced. Only by simulating both fluvial and diffusive transport mechanisms can the model correctly simulate the observed soil distribution. Even though simulating both fluvial and diffusive sediment transport is common practice in many landscape-evolution models (*Tucker and Hancock*, 2010), the interaction between them is often uncertain (*Hancock et al.*, 2002), particularly under unique circumstances like in this field-site: fine-grained aeolian-dominated soils with high degree of temporal variation in soil supply.

Cohen et al., (2015) used the mARM5D model to investigate the differences between bedrock and aeolian dominated soilscape evolution. While the results of that study cannot be directly compared to the results presented in this paper (due to differences in simulation domains and parameterization), they help support some of the assertions proposed by this study. The results in this paper show that different parts of the hillslopes tend to be dominated by one of the two transport mechanisms; diffusion at the top and fluvial at the bottom. Cohen et al., (2015) found an opposite trend for bedrock

weathering dominated soilscapes. In bedrock weathering dominated soilscapes heterogeneity in PSD along the soil profile and selective entrainment by overland flow will result in less erosive (armoured) surface in response to increasing fluvial rates. Therefore increases in runoff rates downslope will not yield a considerable increase in sediment transport (rather an increasingly coarse, source-limited, surface). In aeolian dominated soilscapes the absence of such a mechanism means that increasing runoff downslope will, in the absence of other factors, result in increasing fluvial transport rates downslopes

hence the dominance of this transport mechanism at the bottom part of aeolian dominated hillslopes.

In Cohen et al. (2015) we have also found that heterogeneity in soil distribution, derived from dominance of one of the two sediment transport mechanism, were considerably more pronounced in aeolian-dominated soilscapes. Generally, fluvial sediment transport in bedrock weathering dominated soilscapes, given enough time and pedoturbation stability, will lead to a source-limited flow regime which is in equilibrium with soil production rates (and thus soil depth; Cohen et al., 2015). This

helps explain the patchy soil distribution in aeolian-dominated soilscapes and support our assertion in Cohen et al. (2015) that aeolian dominated soilscapes are more responsive (susceptible) to environmental changes.

Diffusion rate for this field site was found (based on an extensive parametric study) to have a weak and nonlinear relationship with topographic slope ($\beta$=0.1 in Eq. 4). Similarly, the relationship between contributing area and discharge (Eq. 2) was found to be a weak and nonlinear relationship ($n_4$=0.1). These differ from the usual assumption of linearity and may

be due to the aeolian characteristics of the site's soilscape (fine PSD, absence of armouring mechanism, etc.) or, more likely, may be attributed to the relatively steep and concave-down (increase in gradient downslope) characteristics of this field site. Concave-down hillslope means that slope gradients are highest at the lower parts of the hillslope. A linear relationship between slope and diffusion rate, in conjunction with increasing fluvial rates with contributing area, will therefore not allow

for soil to accumulate at the lower part of the hillslope, as observed in our field site. Catena shaped hillslopes can be assumed to have a linear relationship, as slopes are lower at the footslope and toeslope. This will be the focus of a future study.

The overarching goal of this study is the investigation of the history of this region's soilscape evolution in the context of anthropogenically and climatologically driven soil depletion from hillslopes in Europe and southern Israel. The results demonstrate that soil distribution in our field site are highly susceptible to short environmental change. This is in contrast to

the results of Cohen et al. (2013) which demonstrated that the response of bedrock weathering dominated soilscapes to climatic shifts will be a slow transition toward a new steady-state conditions. From this we can again conclude that aeolian dominated soilscapes are more susceptible to environmental change and thus even a relatively small change will lead to considerable alterations. The sharp increase in soil erodibility in the P3 stage representing anthropogenic activity have lead to the good agreement between our simulation results and the observed soil depth distribution at the field. This suggests that

a) soil cover in our field site's hillslopes might have been more extensive during the early Holocene, and b) anthropogenic activity could have led to the soil-depleted hillslopes we observe today.

Soilscape evolution and distribution is typically calculated as a function of landscape topography (as was done here). However, we may also think of an opposite interaction, the effect of soil distribution on landscape evolution (i.e. topography) as studied by *Roering* (2008) on a soil-mantled landscape. The concave shape of the hillslope in this site may be

linked to soil thickening downslope. Thicker soil in this soil-depleted landscape is likely to increase rock weathering (following the 'hump' weathering rate concept), which in this site (limestone dominated lithology) is mostly dissolution, leading to topographic lowering. This process is evident in the micro-topography along the hillslopes (likely propagated by soil/vegetation patches) and from *Saco et al*. (2007) which showed that soil/vegetation bands alter landscape morphology in water-limited environments. This suggests that the massive influx of aeolian sediment during the late-Pleistocene and early-

Holocene may have considerably altered the morphology of this landscape, suggesting a strong coupling between soil production and landscape evolution, a hypothesis that needs to be further investigated.

**5. Conclusions**

Fluvial and diffusive sediment transport mechanisms lead to distinctively different soilscape evolutionary paths. Under

erosive conditions - when transport rates are higher than sediment supply rates - the fluvial mechanism resulted in downslope thinning in soil depth while the diffusion led to downslope thickening. Neither mechanism was able to produce a soil distribution corresponding to that observed in our field-site. Only when both fluvial and diffusive sediment transport

mechanisms were modeled, a reasonable correspondence was achieved. While soilscapes are generally thought off as resulting from both transport mechanisms, this and previous studies demonstrates that fluvial-diffusion coupling is more pronounced in aeolian dominated soilscape.

The results also point to soil distribution features that are indicative of the different sediment transport mechanism. It suggests that, for our semiarid aeolian-dominated field-site, diffusive transport is the dominant mechanism at the top part of the hillslope and fluvial processes are dominant at the. This is in contrast to bedrock weathering dominated soilscapes in which the opposite was observed as increased surface armouring downslope reduces fluvial transport.

Temporal variability in external drivers was shown to be a significant factor in this site's soilscape evolution. This demonstrates the importance of explicitly accounting for geomorphic processes and temporal variability in environmental and anthropogenic dynamics in order to understand the soilscape history, particularly in highly pedogenetic, anthropogenic and climatologically dynamic regions.

This study advanced our understanding of this region's soilscape evolution by elucidating the sediment transport mechanism that may have led to the soil distribution we observe today. The results suggest that relatively swift environmental changes in the late Holocene (i.e. anthropogenic activity) could have considerably changed the site's soil distribution from a soil-mantled hillslopes (albeit with thin loess apron in many locations) to the soil-depleted hillslopes we observe today. Additional research is needed (and is ongoing) to better confine the rates and spatiotemporal dynamics of soil erosion and development in this site.

### Acknowledgments

This research was supported by the Israel Science Foundation (ISF) (grant 1184/11). GRW was supported by an Australian Research Council Australian Professorial Fellowship.

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

**Table and Figure captions**

Table 1. Values of the parameters that are driven by the simulation scenario. P1, P2 and P3 are the three simulated periods (80-12, 12-8 and 8-0 kyr BP respectively; Figure 3) and S1, S2 and S3 are the three simulation (Fluvial, Diffusive and Combined respectively).

|  | *i*: Erodibility, *e* (unitless; Eq. 1) | *ii*: Runoff, *Q* (m³/y; Eq. 2) | *iii*: Aeolian deposition, $K_a$ (mm/y; Eq. 5) | *iv*: Diffusive rate, $D_o$ (mm/y; Eq. 3-4) |
|---|---|---|---|---|
| P1 | S1: 0.0001 | S1: 0.0017 | S1: 0.1 | S1: 0.0 |
|  | S2: 0.0 | S2: 0.0 | S2: 0.1 | S2: 21.5 |
|  | S3: 0.0001 | S3: 0.00066 | S3: 0.1 | S3: 12.0 |
| P2 | S1: 0.0002 | S1: 0.0034 | S1: 0.05 | S1: 0.0 |
|  | S2: 0.0 | S2: 0.0 | S2: 0.05 | S2: 21.5 |
|  | S3: 0.0002 | S3: 0.00132 | S3: 0.05 | S3: 12.0 |
| P3 | S1: 0.001 | S1: 0.017 | S1: 0.01 | S1: 0.0 |
|  | S2: 0.0 | S2: 0.0 | S2: 0.01 | S2: 10.75 |
|  | S3: 0.001 | S3: 0.0066 | S3: 0.01 | S3: 6.0 |

Figure 1. The study region and site. The Long Term Ecological Research, LTER, in southern Israel, is uniquely situated at a margin between Mediterranean climatic regime to the north and arid climate to the south (a). The study site is located on a Loess belt (b) deposited during the late-Pleistocene and early-Holocene. Hillslopes in this region are today mostly depleted of their Loess cover (c and d). The contour lines in (c) are at 2 m intervals.

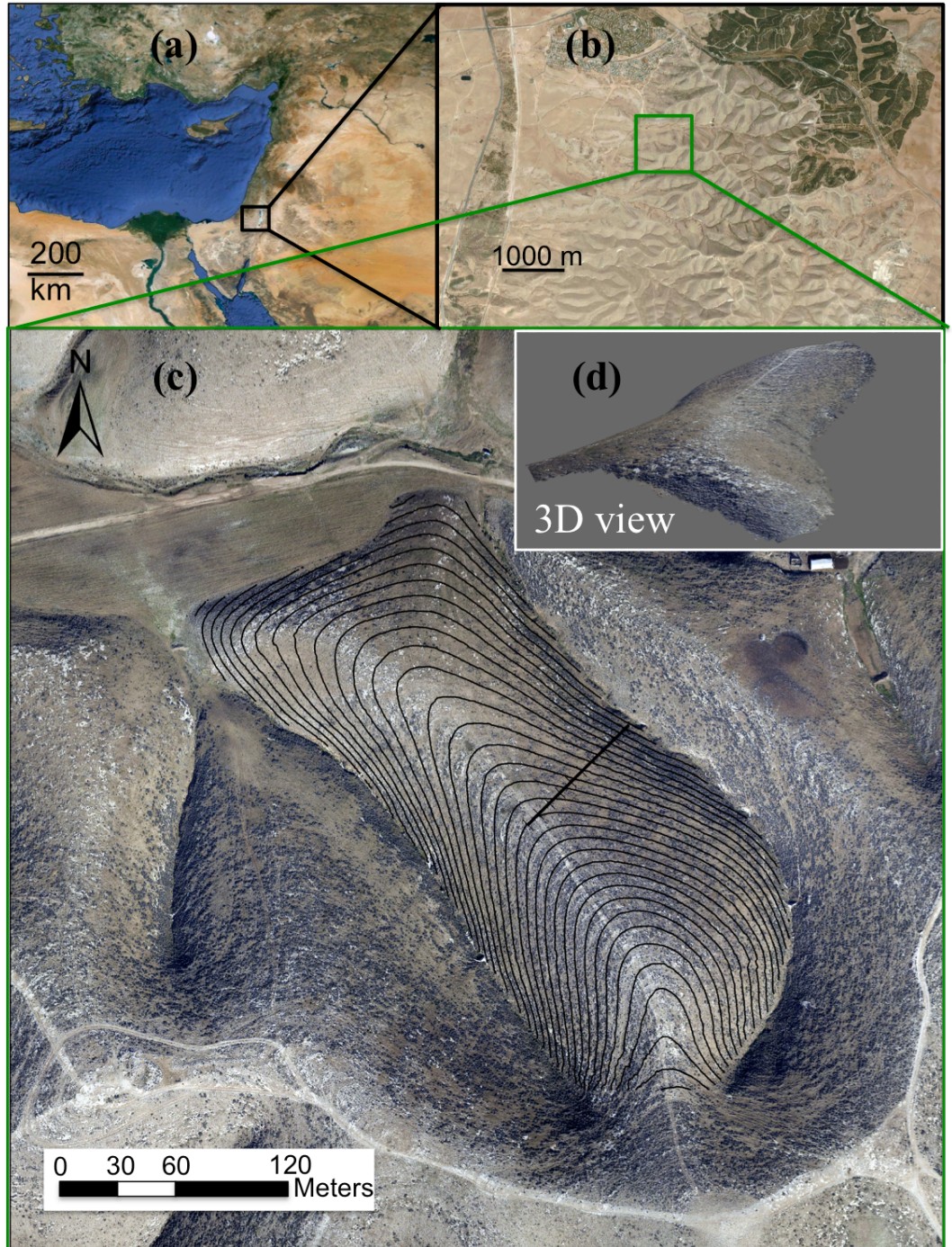

Figure 2. Soil depth at the Lehavim LTER site, measured in 350 locations and interpolated using Kriging. Pixels classified as rock from a 10 cm² orthophoto were assigned zero depth. The contour lines are at 2 m intervals and the thick black line is the location of the transect analyzed in section 3.3.

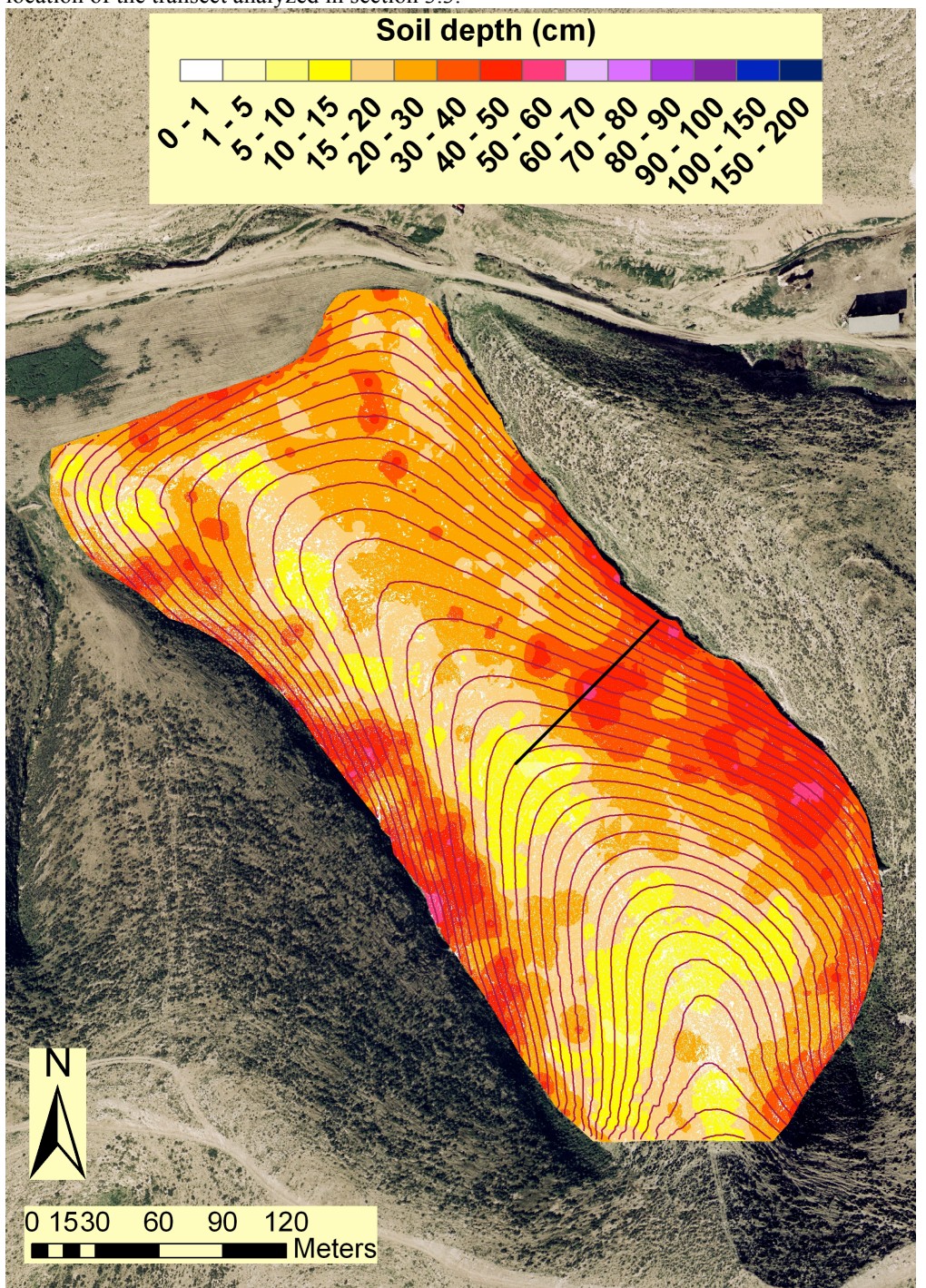

Figure 3. The simulation scenario. Describe the temporal changes in four model parameters as a function of climatic and anthropogenic drivers. The Erosion factor is overlapping with the Runoff generation line.

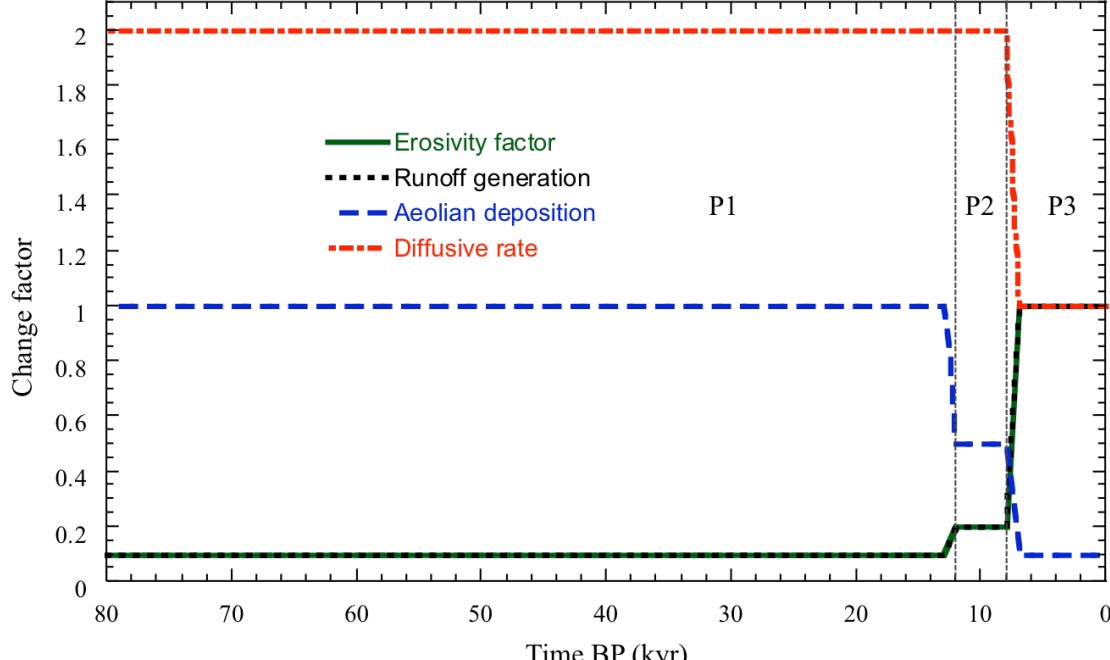

Figure 4. Soil depth maps produced by the Fluvial only simulation (S1) at 3.2 kyr time intervals in the last 18 of 80 kyr simulated: (a) 18 kyr BP (b) 12.8 kyr BP (end of P1); (c) 9.6 kyr BP; (d) 6.4 kyr BP (end of P2); (e) 3.2 kyr BP; and (f) 0 kyr BP (final/modern- end of P3). The contour lines represent 2 m change in topography.

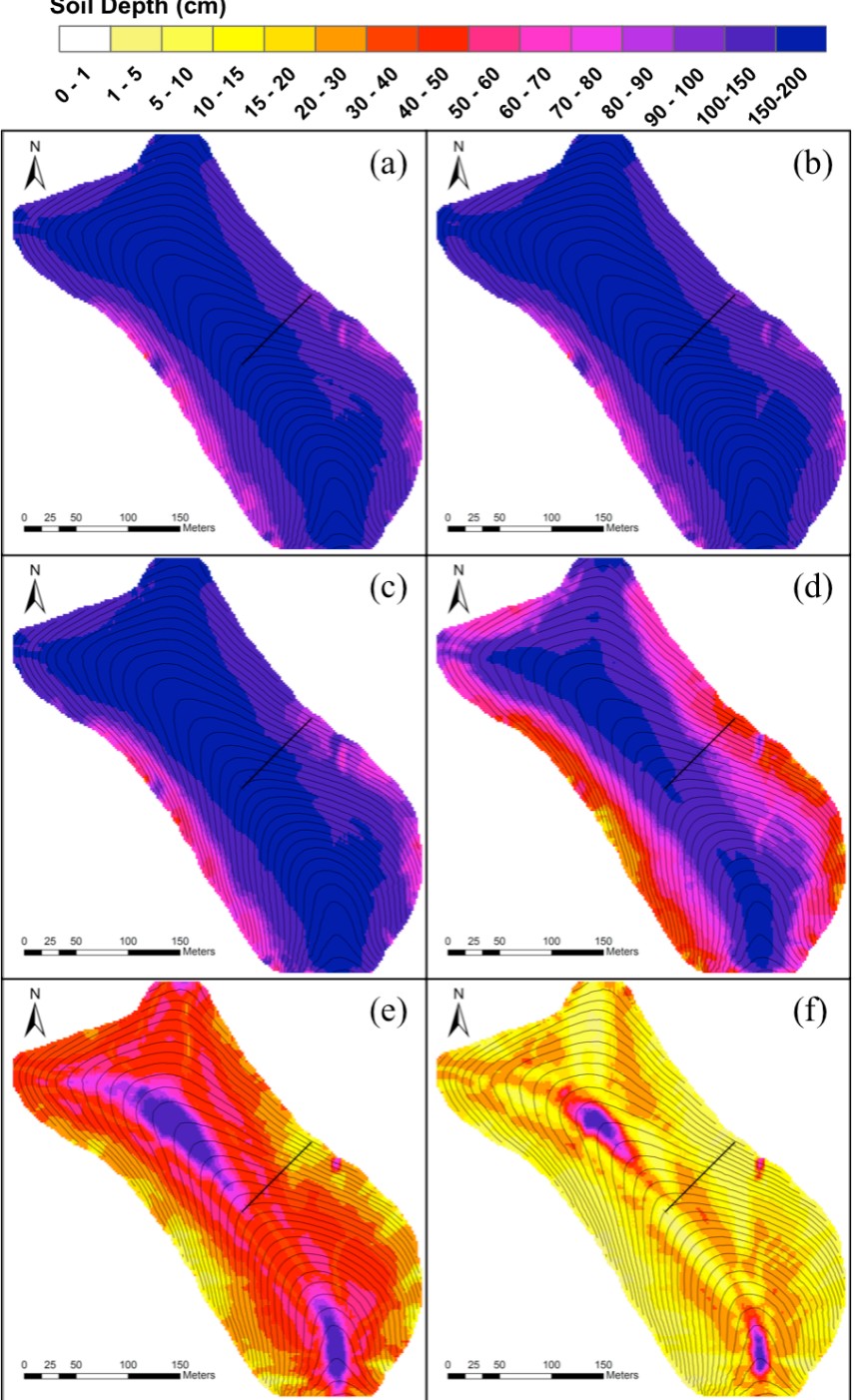

Figure 5. Soil depth maps produced by the Diffusive only simulation (S2) at 3.2 kyr time intervals in the last 18 of 80 kyr simulated: (a) 18 kyr BP (b) 12.8 kyr BP (end of P1); (c) 9.6 kyr BP; (d) 6.4 kyr BP (end of P2); (e) 3.2 kyr BP; and (f) 0 kyr BP (final/modern- end of P3). The contour lines represent 2 m change in topography.

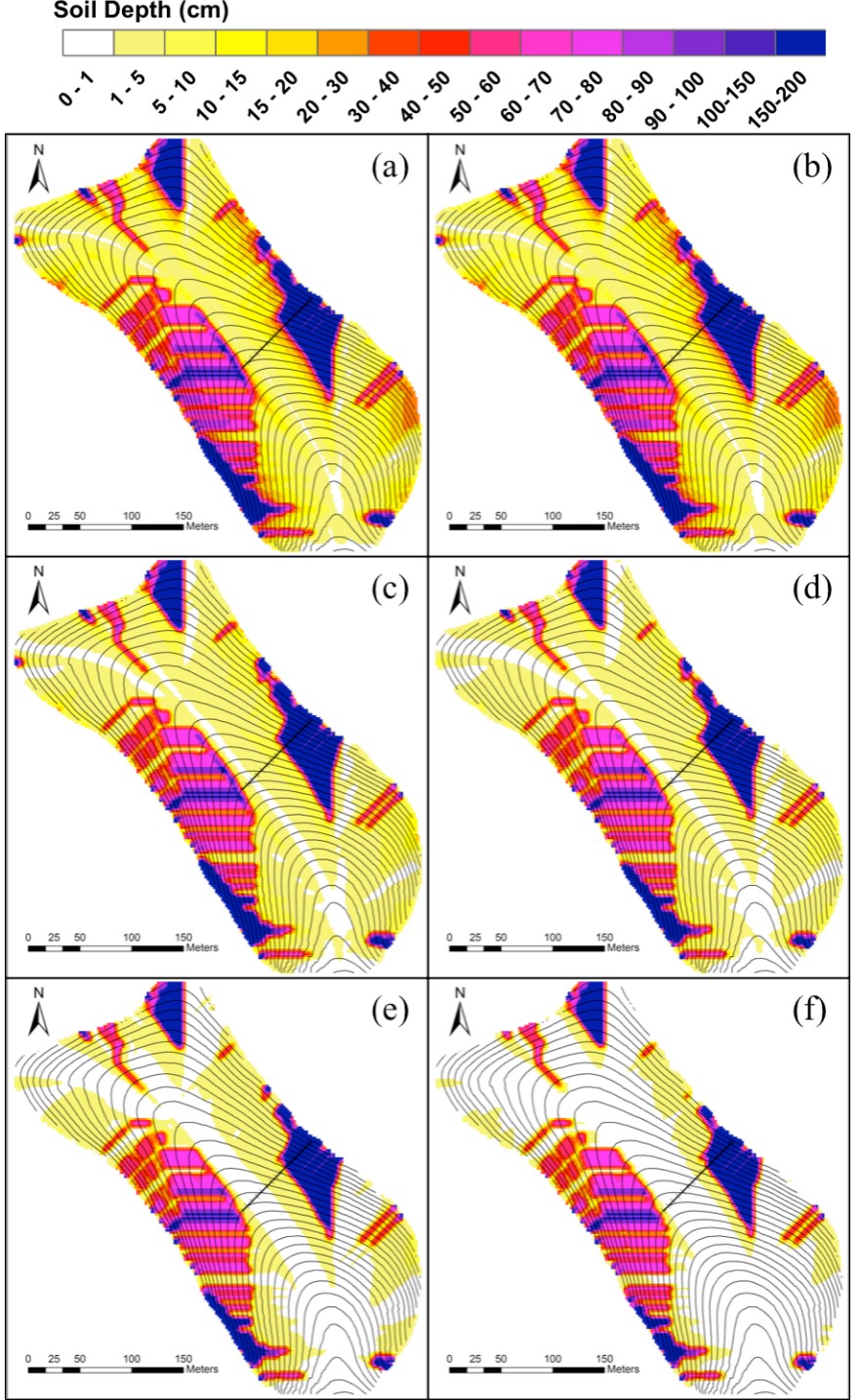

Figure 6. Soil depth maps produced by the Combined simulation (S3) at 3.2 kyr time intervals in the last 18 of 80 kyr simulated: (a) 18 kyr BP (b) 12.8 kyr BP (end of P1); (c) 9.6 kyr BP; (d) 6.4 kyr BP (end of P2); (e) 3.2 kyr BP; and (f) 0 kyr BP (final/modern- end of P3). The contour lines represent 2 m change in topography.

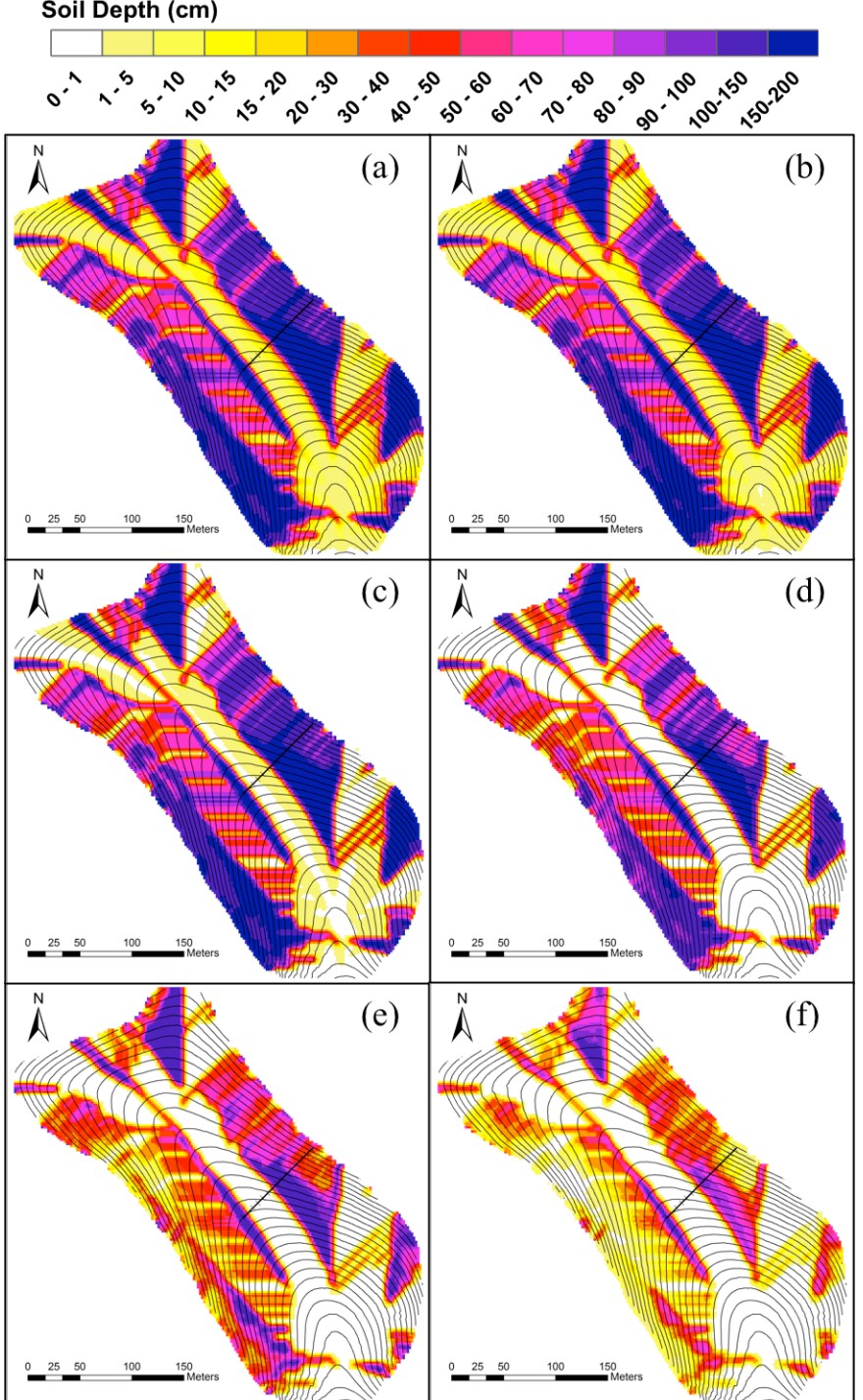

Figure 7. Soil depth across a transect (Figure 1c) on the northeastern facing hillslope at 6 time intervals (corresponding to soil maps in Figures 4-6): (a) the Fluvial only simulation (S1), (b) the Diffusive only simulation (S2), (c) the Combined simulation (S3), and (d) measured soil depth (Figure 2).

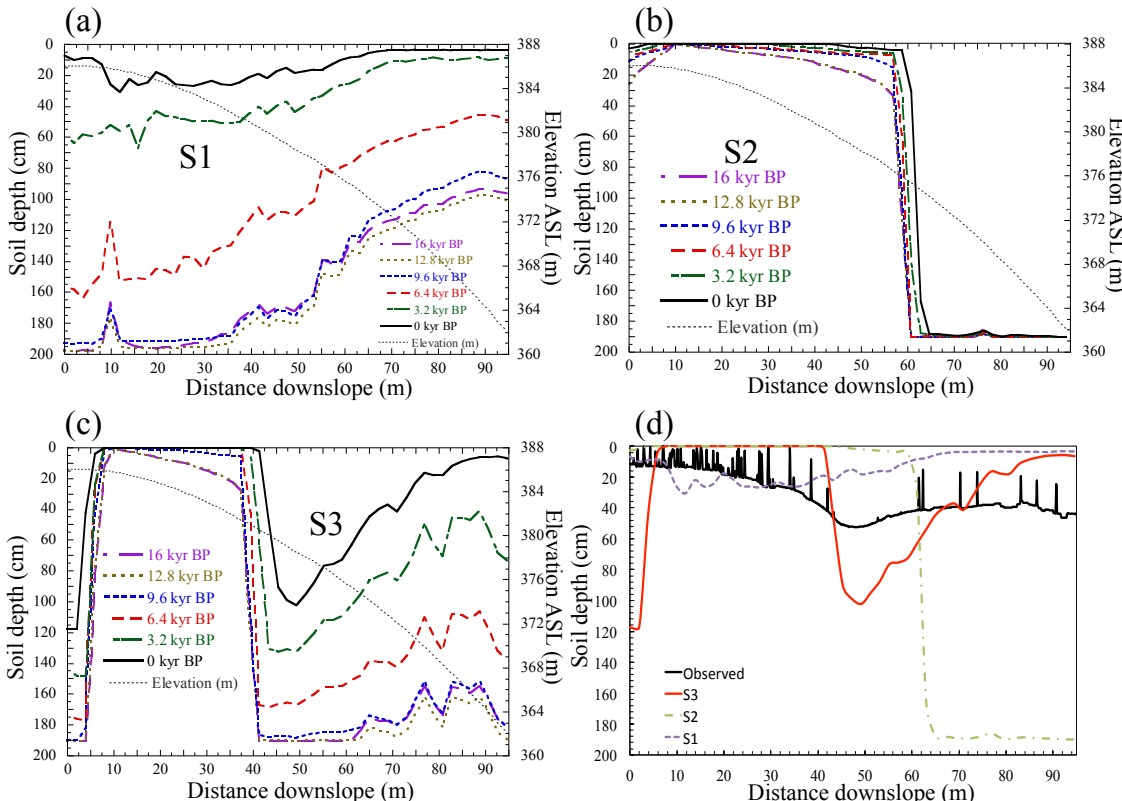

