# Peer review of "Soilscape evolution of aeolian-dominated hillslopes during the Holocene: investigation of sediment transport mechanisms and climatic-anthropogenic drivers"

_Earth Surface Dynamics, 2016_

## Referee Comment (RC1) · Anonymous Referee #1 · 6 Feb 2016

This paper describes the application of a coupled pedogenic-geomorphic model to a semi-arid field site in Israel. The authors demonstrate that a model that combines transport by diffusive hillslope processes (creep and bioturbation) with transport by overland and rill flow does a better job at reproducing observed soil depths than a model that contains just one of these transport types. The paper concludes that different parts of the hillslopes tend to be dominated by one of the two transport mechanisms: diffusion at the top and fluvial at the bottom.

That a model that includes both fluvial and diffusive processes works better than one

with just diffusion or just fluvial processes does not strike me as a significant conclusion. I don't understand why the authors would run simulations with diffusive processes only or fluvial processes only, given that all landscapes clearly have both of these transport types occurring.

Model concerns:

Equation 1: Nothing like equation 1 appears in Engelund and Hansen (1968). In sediment transport the flux usually goes as the square root of the excess density, not the square of excess density (which is usually s-1, not 1-s as shown here). The assumption that sediment flux is linear with water discharge (i.e. $n\_1 = 1$) is inconsistent with all sediment transport formula in the literature. The publication year is 1967, not 1968, and Engelund's name is misspelled.

Equation 2: Why do the authors assume $n\_4 = 0.1$? No reason is given. This is an extremely low scaling relationship between discharge and area.

Routing: D8 routing is inappropriate for hillslopes. The authors need to use a multiple flow direction algorithm. As the authors state, the unrealistic "striping" of the model output along 45 deg angles is a result of the routing algorithm. However, it is very strange that the striping occurs only for the diffusive simulation, which does not involve routing at all as far as I can determine from the text.

Equations 3&4: This does not look like hillslope diffusion. The authors have assumed that the colluvial transport rate increases as the 0.1 power of slope, not the usual linear formulation (or the nonlinear formulation of Bucknam and Andrews, Roering et al., etc.). No results of the calibration used to obtain beta = 0.1 is provided. The units of the various parameters are very hard to keep track of and clearly wrong in some cases. $D\_s$ should not have units of time because the time step is included in equation 4 ($D\_s$ should have units of m, not m/yr). What is k? What are its units and its value?

Equation 5: The reader is referred to Minasny and McBratney (2006), which is not

in the reference list. When I tracked down Minasny and McBratney (2006) I found a rather different equation (their equation (4)). Equation (5) is dimensionally incorrect. It is wrong to have the steady state weathering rate appear inside the exponential – the argument of any exponential should be unitless. Why are delta_1 and delta_2 equal to 4 and 6? What are the units? If they are meters these are very large values (i.e. they imply that weathering rates fall off by a factor of e only once the soil is at least 4 m thick. This is a very thick soil).

How the equations are combined is not clear. There must be some conservation equation being used in the model (e.g. erosion rate is related to the divergence of sediment flux), but this is not shown. I did find something like a conservation equation in Cohen et al. (2015), but that equation is dimensionally incorrect (the erosion rate (which has units of L/T) is equated with sediment flux, which has units of L^2/T).

Fluvial erosion from hillslopes is generally modeled as a 2-step process: 1) rainsplash disturbance of soil aggregates to liberate them into the water column and 2) size-selective transport. Only the second process is considered in this model.

The model does not include the vertical redistribution of aeolian material (aeolian deposits stay on the surface). In nature, the reason why an argillic horizon forms is that aeolian fines are redistributed downward in the soil profile. Therefore, I don't see how this is a realistic model for pedogenesis.

Calibration concerns:

Some of the model parameters are chosen ad hoc (i.e. beta = 0.1, n_4 = 0.1) with no apparent calibration. Some are simply chosen based on the default values in other studies that may or may not be realistic for the study site in Israel. No data were used to relate climate changes to the model parameters. The "change factor" values and how they were modified over time may be qualitatively correct but the absolute values appear to be ad hoc. Some data must be used for calibration.

More broadly, the model has so many parameters (I lost count – a table of parameters, their units, and their chosen values would have helped) that I cannot see how a search of the parameter space could possibly have been done to find the optimal values, except via a Bayesian approach such as MCMC. Calibrating a model with 10 or 20+ parameters to a dataset that constrains only one element of the system (soil depth in this case) has to be done very carefully if it can be done at all. In cases where model parameters were matched to the observed data using an "extensive parametric study", no details are provided. This makes it very difficult to have confidence in the conclusions. When the model "fails" to match the data for the fluvial case or the diffusive case, perhaps it is simply that the model hasn't been properly calibrated.

The paper concludes that different parts of the hillslopes tend to be dominated by one of the two transport mechanisms; diffusion at the top and fluvial at the bottom. I don't see how the numerical experiments support the conclusion that fluvial processes dominate at the bottom. This conclusion is inconsistent with Tarboton et al. (1992) and many more recent studies (Perron et al., 2008; 2009) that conclude that the transition from diffusive to fluvial dominance occurs at the channel head. As long as one is on an unincised hillslope, diffusive processes should be dominant everywhere according to the published literature. The discrepancy between the results of this paper and previous studies could simply be a result of the very unrealistic value of beta (0.1) chosen with no justification. Certainly fluvial erosion must become relatively more important at the base of the slopes compared to the top because the contributing area goes to zero at the top (hence the importance of fluvial transport must go to zero at the divide). However, I do not see any evidence in the paper that fluvial processes dominate diffusive processes at the base of the slope.

Other concerns:

The authors state that soil development is dominated by aeolian processes at their study site but no evidence is provided to demonstrate this. There are now a number of studies that use immobile elements (Ti, Zr) to quantify the relative dominance of

aeolian input versus in situ weathering of parent material in soils. So, it may be possible to constrain this yet I don't see how this was done in this study. Similarly, the key motivating question of the study is whether soil degradation is caused by climate change or anthropogenic forcing. I don't see any evidence in this paper that soils were depleted (i.e. that they were thicker and have now thinned). Moreover, the question of whether soil degradation occurred by climate change or human activity relates primarily to the timing of the soil degradation. There is no geochronology or other evidence presented to address this question. These are two examples of many in which facts were assumed about the study site without evidence.

I don't understand why the observed soil depth (Fig. 7D) shows "spikes" in the plot. The color map from Fig. 2 does not show these spikes in the data.
* * *

---

## Referee Comment (RC2) · Anonymous Referee #2 · 24 Feb 2016

This paper presents a simulation of soil-landscape evolution in a semiarid zone of Israel under fluvial and diffusive sediment transport.

The paper is nicely written and the method is clearly described. The results are stimulating, offering possible soil-landscape evolution pathways.

My comments are that some of the parameters are arbitrarily chosen, e.g. the humped model of weathering. While theoretically it is sound, but no published result yet showing such weathering parameters. The authors wrote: "Limestone bedrock typically results in limited soil production by weathering except for producing a Mollisol" The statement

on limited production and Mollisol is not necessary true, Mollisol development is due to accumulation of loess and organic matter.

Another limited assumption is aeolian deposition which is uniform, what are the particle sizes of the aeolian deposit? Silt-size? 2-20 um?

The simulation is run for 16,000 years. It would be beneficial to see how much of the "soil" is due to bedrock weathering and how much is due to aeolian deposit. There is no mention of vegetation effect? Is there a possible feedback between vegetation and erosion?

---

## Author Response (AR1)

*This paper describes the application of a coupled pedogenic-geomorphic model to a semi-arid field site in Israel. The authors demonstrate that a model that combines transport by diffusive hillslope processes (creep and bioturbation) with transport by overland and rill flow does a better job at reproducing observed soil depths than a model that contains just one of these transport types. The paper concludes that different parts of the hillslopes tend to be dominated by one of the two transport mechanisms: diffusion at the top and fluvial at the bottom.*

First of all we would like to thank the referee for this very thorough review. Below we provide a point-by-point response to the reviewer's comments.

*That a model that includes both fluvial and diffusive processes works better than one with just diffusion or just fluvial processes does not strike me as a significant conclusion. I don't understand why the authors would run simulations with diffusive processes only or fluvial processes only, given that all landscapes clearly have both of these transport types occurring.* The reason why is clearly and repeatedly described and discussed. In a nutshell: the relationship between these transport types are well explored in bedrock-weathering dominated but not in aeolian soilscapes. Moreover, we investigate the effects of temporal variability in external drivers (i.e. climatic/anthropogenic scenarios, a key aspect of the overarching research) that seems to, based on our literature review for this region, differ for the two transport mechanisms. We therefore need to isolate each mechanism in order to identify their specific spatial and temporal dynamics. The conclusion of this study is NOT that both transport mechanisms are in play but rather a suite of insights on their specific dynamics and interactions.

*Model concerns:*

*Equation 1: Nothing like equation 1 appears in Engelund and Hansen (1968). In sediment transport the flux usually goes as the square root of the excess density, not the square of excess density* We disagree with this statement (see page 48 in Engelund and Hansen (1968)) but acknowledge that we should have been more specific. This equation was based on the TOPOG model sediment transport calculations which used the Engelund and Hansen (1968) equation ([http://www-data.wron.csiro.au/topog/user/contents/frame1.0.html](http://www-data.wron.csiro.au/topog/user/contents/frame1.0.html)). This is now clearly outlined in the revised manuscript.

*(which is usually s-1, not 1-s as shown here).* True, this was a typo but make no difference to the result as the expression is squared and s is constant.

*The assumption that sediment flux is linear with water discharge (i.e. $n\_1 = 1$) is inconsistent with all sediment transport formula in the literature.* The n1 and n2 values are in line with the TOPOG values.

*The publication year is 1967, not 1968, and Engelund's name is misspelled.* Corrected.

*Equation 2: Why do the authors assume n_4 = 0.1? No reason is given. This is an extremely low scaling relationship between discharge and area.* It is explained: *"In Cohen et al. (2010 and 2015) the relationship between contributing area and runoff discharge was assumed to be linear (n4=1). This assumption could not be justified in our field-site as Yair and Kossovsky (2002) showed that runoff generation in this region does not increase linearly downslope. We therefore use n4=0.1 in this study."*

*Routing: D8 routing is inappropriate for hillslopes. The authors need to use a multiple flow direction algorithm. As the authors state, the unrealistic "striping" of the model output along 45 deg angles is a result of the routing algorithm.* This limitation is discussed in previous papers. One of the main advances in the mARM model is its computational efficiency which allowed, for the first time, for such explicit and long-term simulations at landscape scales. Recent advances with this framework and similar models may now allow us to consider more runtime-expensive routing algorithms. This is a topic for a whole different paper.

*However, it is very strange that the striping occurs only for the diffusive simulation, which does not involve routing at all as far as I can determine from the text.* Diffusion absolutely involved routing, how else does the model transport the sediment down the slope? It is, however, not directly affected by contributing area and so there is no downstream scaling and the stripping occurs along the hillslope in places where the flow directions are parallel.

*Equations 3&4: This does not look like hillslope diffusion.* This stem from our attempt to simplify a complex 5D (x, y, z, t and PSD) algorithm in a couple of equations. As we describe, this is a novel equation and is described and explored in more details in Cohen et al. (2015). It is now better explained in the revised manuscript.

*The authors have assumed that the colluvial transport rate increases as the 0.1 power of slope, not the usual linear formulation (or the nonlinear formulation of Bucknam and Andrews, Roering et al., etc.). No results of the calibration used to obtain beta = 0.1 is provided.* In Cohen et al. (2015) we used a linear relationship (b=1) and have conducted an extensive parametric study for beta for this study. See the relationship between slope and diffusion rate for beta = 0.1 and 1 in the plots bellow. The fact that beta differs from 1 to such an extent in this field-site is actually very interesting! We propose that it could be another aeolian-driven affect (fine PSD, absence of armoring etc.) or is due to the relatively steep and concave down shape of the hillslopes in this site (so site-specific). This is now explained (in the model description) and discussed (in the discussion section) in the revised manuscript.

[Figure]

[Figure]

*The units of the various parameters are very hard to keep track of and clearly wrong in some cases. D_s should not have units of time because the time step is included in equation 4 (D_s should have units of m, not m/yr).* Not true! Equation 4 doesn't have a time step parameter, it describes changes in D (m/s) down the profile. In practice it makes not difference.

*What is k? What are its units and its value?* "…k is the surface diffusivity (m)" its values are described in section 2.4. We changed k to $D_o$ to distinguish it from $k_a$.

*Equation 5: The reader is referred to Minasny and McBratney (2006), which is not in the reference list.* Corrected

*When I tracked down Minasny and McBratney (2006) I found a rather different equation (their equation (4)). Equation (5) is dimensionally incorrect. It is wrong to have the steady state weathering rate appear inside the exponential – the argument of any exponential should be unitless.* The equation was indeed (as stated) modified, the Po parameter was moved to allow for an above-zero watering rate at the surface while depth rates down the soil profile asymptote to zero. Cohen et al. (2010) focused on the model weathering equations and algorithm. We can see how the description of the model weathering calculations is confusing and misrepresentative. This stems, again, from our attempt to simplify a complex algorithm into one equation with time-varying parameter. The full algorithm includes compiling a transition matrix which control the transition of PSD in each particle size class to smaller class(s) in each soil-profile layer. The algorithm was described at length in Cohen et al. (2009 and 2010). The depth-varying weathering rate equation in the model (see the actual model FORTRAN function below) is not directly used to calculate weathering rate, rather the relative (normalized) change in weathering down the profile. As part of the revised model description we removed the section describing the weathering calculations as it does not include parameters that are modified in the simulation scenarios we analyzed in this paper.

```
!####################################################################
! A function to calculate the decline in weathering rate as a function of depth
! It return the WeatheringAlpha which is different for every layer
! The exponential function is taken from Minasny & McBratney (2006)
! de/dt=Po{Exp(-k1h)-Exp(-k2h)}+Pa ;
! Their original values are: Po-potential WR=0.25; k1=4; k2=6; Pa- steady state WR=0.005
! The function was changed in version 4.5.1 to account to close to zero weathering in
lower layers
!####################################################################
    REAL*8 Function DepthWeatheringAlpha(WeatherAlpha,i,LayerDepth)
    IMPLICIT NONE
```

```
    Integer i
    Real*8 WeatherAlpha, LayerDepth, Ratio, Depth
    If (i==1) Then
    Depth=0.5
    Else
    Depth=(i-1)*LayerDepth-(LayerDepth/2)
    End If
    Depth=Depth/100 !convert to meters
    Ratio=(0.25*((EXP(-4*Depth+0.02))-(EXP(-6*Depth))+0))/0.04 !We divide by 0.04 to
normalize it
    DepthWeatheringAlpha = WeatherAlpha*Ratio
    Return
    End Function DepthWeatheringAlpha
```

*Why are delta_1 and delta_2 equal to 4 and 6? What are the units? If they are meters these are very large values (i.e. they imply that weathering rates fall off by a factor of e only once the soil is at least 4 m thick. This is a very thick soil).* Following on the comment above, the model algorithm uses the Minasny and McBratney (2006) equation to get the relative change in weathering rate down the soil profile. These variables therefor control the shape rather then the actual weathering rate. Their values were based on Minasny and McBratney (2006) to maintain the relative change (i.e. shape of the hump function) and so in reality they are unitless.

*How the equations are combined is not clear. There must be some conservation equation being used in the model (e.g. erosion rate is related to the divergence of sediment flux), but this is not shown. I did find something like a conservation equation in Cohen et al. (2015), but that equation is dimensionally incorrect (the erosion rate (which has units of L/T) is equated with sediment flux, which has units of L^2/T).* Again, as we showed above, there is no dimensionality issue with the actual implementation of the equation. As we stated at the start of section 2.2 the full description of the model architecture is provided in Cohen et al. (2009 and 2010). The model description in this paper is limited to the equations "that include the parameters that are modified by the simulation scenarios we analysed here." A short description of the model architecture was added to the revised manuscript.

*Fluvial erosion from hillslopes is generally modeled as a 2-step process: 1) rainsplash disturbance of soil aggregates to liberate them into the water column and 2) size-selective transport. Only the second process is considered in this model.* Not exactly, the sediment transport process is lumped. We argue that this is appropriate for the scales (primarily temporal) we simulate. Soilscape evolution is extremely complex and numerical models cannot, and are not intended to, exactly and fully mimic it. Models are useful for simplifying complex dynamics, allowing us to isolate parameters and processes and test hypotheses and concepts. Modeling results must be interpreted within the model assumptions, a concept that we carefully follow.

*The model does not include the vertical redistribution of aeolian material (aeolian deposits stay on the surface). In nature, the reason why an argillic horizon forms is that aeolian fines are redistributed downward in the soil profile. Therefore, I don't see how this is a realistic model for pedogenesis.* Same comment as above, consider the scales and focus of this study.

*Calibration concerns:*

*Some of the model parameters are chosen ad hoc (i.e. beta = 0.1, n_4 = 0.1) with no apparent calibration. Some are simply chosen based on the default values in other studies that may or may not be realistic for the study site in Israel. No data were used to relate climate changes to the model parameters. The "change factor" values and how they were modified over time may be qualitatively correct but the absolute values appear to be ad hoc. Some data must be used for calibration.* Expending on our previous comments, the goal of this study is not to precisely predict soil dynamics but rather to isolate and conceptually analyze specific processes and dynamics. That can only be done with numerical models given the complexity and longevity of many of the processes involved. Over the years we have used best available data to calibrate some of the model parameters. In a limited and qualitative ways this is what we have done here with observed soil distribution. However we have found that using observed data to calibrate a specific model parameter is extremely problematic as, almost always, the parameter dynamics (spatial and temporal) cannot be sufficiently isolated form the observations. For this field site we actually have quite detailed paleoclimate and hydrological data (references are provided in the manuscript) but we intestinally simplified it. We did so for two reasons: (1) it allows for a much clearer analysis and (2) we are not attempting to precisely predict soilscape dynamics. This again relate to our previous comment about the use of models for soilscape evolution studies.

*More broadly, the model has so many parameters (I lost count – a table of parameters, their units, and their chosen values would have helped) that I cannot see how a search of the parameter space could possibly have been done to find the optimal values, except via a Bayesian approach such as MCMC. Calibrating a model with 10 or 20+ parameters to a dataset that constrains only one element of the system (soil depth in this case) has to be done very carefully if it can be done at all. In cases where model parameters were matched to the observed data using an "extensive parametric study", no details are provided. This makes it very difficult to have confidence in the conclusions. When the model "fails" to match the data for the fluvial case or the diffusive case, perhaps it is simply that the model hasn't been properly calibrated.* See comment above, these issues were address in the last four papers about this model.

*The paper concludes that different parts of the hillslopes tend to be dominated by one of the two transport mechanisms; diffusion at the top and fluvial at the bottom. I don't see how the numerical experiments support the conclusion that fluvial processes dominate at the bottom.* This is explained at the results and discussion sections: each of the transport mechanisms resulted (when simulated alone) in fairly distinct soil dynamics. When the two were simulated together their signatures were visible in different parts of the soilscape.

*This conclusion is inconsistent with Tarboton et al. (1992) and many more recent studies (Perron et al., 2008; 2009) that conclude that the transition from diffusive to fluvial dominance occurs at the channel head. As long as one is on an unincised hillslope, diffusive processes should be dominant everywhere according to the published literature.*

Good, now you are getting to a main goal of this paper – investigating the potential differences between aeolian and bedrock ("normal") soilscape evolution. These comments support our assertion and results that the two are indeed different - though the references and your comment focus on channel-hillslope interaction and we are only looking at hillslope processes here. We admit that the use of the term 'fluvial' is confusing in this context but we explicitly explain this in the introduction. This is stated and discussed in the manuscript, even in the context of our previous study: "In Cohen et al., (2015) we found an opposite trend for bedrock weathering dominated soilscapes." (section 4. Discussion).

*The discrepancy between the results of this paper and previous studies could simply be a result of the very unrealistic value of beta (0.1) chosen with no justification.* As stated above we did provide justification to beta. See comment above about equation 4.

*Certainly fluvial erosion must become relatively more important at the base of the slopes compared to the top because the contributing area goes to zero at the top (hence the importance of fluvial transport must go to zero at the divide).* True but this is a highly spatially and temporally dynamic change, not as simple as you describe it.

*However, I do not see any evidence in the paper that fluvial processes dominate diffusive processes at the base of the slope.* This is discussed in the manuscript. In a nutshell: if you compare soil depth evolution between the three simulations (Figures 4-6) and the cross section results (Figure 7) you will see that diffusion led to thick soils at the base in contrast to the fluvial simulation. The combined simulation looks a lot like the fluvial simulation at the base and like the diffusive at the middle part of the hillslope. We acknowledge that this is a qualitative observation but (as commented above about model assumptions and simplifications) we assert that it is most appropriate.

*Other concerns:*

*The authors state that soil development is dominated by aeolian processes at their study site but no evidence is provided to demonstrate this. There are now a number of studies that use immobile elements (Ti, Zr) to quantify the relative dominance of aeolian input versus in situ weathering of parent material in soils. So, it may be possible to constrain this yet I don't see how this was done in this study.* The loess belt in the northern Negev has been well studied; including the rates, extent and dynamics of aeolian deposition in this region (some references are provided in the manuscript). Indeed all one need to do is walk the site and see the extent of loess deposition and how little soil is produced by bedrock weathering. That been said (and repeating an earlier responses), this is actually not a crucial point for this conceptual study.

*Similarly, the key motivating question of the study is whether soil degradation is caused by climate change or anthropogenic forcing. I don't see any evidence in this paper that soils were depleted (i.e. that they were thicker and have now thinned). Moreover, the question of whether soil degradation occurred by climate change or human activity relates primarily to the timing of the soil degradation. There is no geochronology or other evidence presented to address this question. These are two examples of many in*

*which facts were assumed about the study site without evidence.* No, this is the overarching motivation not the goal of this study. The goal of this study is to gain conceptual insights into the soilscape evolution and the impacts of time varying parameters. As described above, we intentionally used broad-brush estimates of climatic and anthropogenic changes. Ongoing research (which will couple field and modeling efforts) is looking into the question of whether or not soil ever accumulated on the hillslopes. These points are clearly stated in the manuscript.

*I don't understand why the observed soil depth (Fig. 7D) shows "spikes" in the plot. The color map from Fig. 2 does not show these spikes in the data.* The soil map (Fig 2) is the product of interpolation between measurement points with exposed bedrock (classified form aerial photography) "burned" as zero depth (white color in Fig 2). This was explained in section 2.1. If you look closely you could see these zero-slope ("spikes") along the transect.

**Anonymous Referee #2**
*This paper presents a simulation of soil-landscape evolution in a semiarid zone of Israel under fluvial and diffusive sediment transport.*

*The paper is nicely written and the method is clearly described. The results are stimulating, offering possible soil-landscape evolution pathways.*

First of all we would like to thank the referee for this thorough review. Below we provide a point-by-point response to the reviewer's comments.

*My comments are that some of the parameters are arbitrarily chosen, e.g. the humped model of weathering. While theoretically it is sound, but no published result yet showing such weathering parameters.* As discussed at length in the response for review #1, it is true (and acknowledged in the manuscript) that many of the model parameters cannot be explicitly and directly calibrated even if extensive field data was available. This stems from the difficulty in isolating specific parameters from field observations, the longevity of the processes simulated and their complex interconnectivity and spatiotemporal dynamics. That is one of the motivations for developing a soilscape evolution model and for this study, isolating processes and parameters, allowing us to develop (and conceptually test) hypotheses on soilscape evolution pathways and drivers. That being said we have, over the last several years, focused our analysis on specific processes and parameters, gaining important insights into some of the model parameters. Most specifically, and relating to this comment, is our extensive analysis of weathering equations and parameters in Cohen et al. (2010). Moreover the hump weathering equation we used was adopted from Minasny and McBratney (2006) which is based on their extensive research.

*The authors wrote: "Limestone bedrock typically results in limited soil production by weathering except for producing a Mollisol" The statement on limited production and Mollisol is not necessary true, Mollisol development is due to accumulation of loess and*

*organic matter.* So why is this not true? We state: "…typically results in…" and go on to explain what is and is not actually simulated.

*Another limited assumption is aeolian deposition which is uniform, what are the particle sizes of the aeolian deposit? Silt-size? 2-20 um?* Good point. The aeolian deposition PSD is the same as the one we used in Cohen et al. (2015) and is shortly described there. We now clarify that in the manuscript: "We use the same PSD as in the fine-grained simulation in Cohen et al., (2015), with a $d_{50}$=0.06 mm." (section 2.1.3).

*The simulation is run for 16,000 years. It would be beneficial to see how much of the "soil" is due to bedrock weathering and how much is due to aeolian deposit.* As describe in section 2.4, bedrock weathering is assumed to be very low (0.01 mm/y), an order of magnitude lower tan aeolian deposition (0.1 mm/y). Also an explicit analysis of the two soil production mechanisms was explored in Cohen et al. (2015).

*There is no mention of vegetation effect? Is there a possible feedback between vegetation and erosion?* Of course, as well with bioturbation, crusts etc. These kind of caveats were discussed quite extensively in previous papers (particularly in Cohen et al., 2015). This fact is now described in the manuscript.

**Main document changes and comments**

| Page 4: Deleted | Sagy Cohen | 3/29/16 10:32 AM |
|---|---|---|

(%)

| Page 4: Inserted | Sagy Cohen | 3/29/16 10:32 AM |
|---|---|---|

(m/m)

| Page 4: Deleted | Sagy Cohen | 3/28/16 11:31 AM |
|---|---|---|

modeling

| Page 4: Inserted | Sagy Cohen | 3/28/16 11:31 AM |
|---|---|---|

modelling

| Page 4: Inserted | Sagy Cohen | 3/28/16 11:32 AM |
|---|---|---|

The mARM framework introduced a novel implementation of physically-based equations using transition matrices that express the relative change in spatially and temporally explicit PSD vectors. This concept greatly improves the model computational efficiency and modularity but is challenging to describe in full. Below we describe the mARM5D physically-based equations that include the parameters that are modified in the simulation scenarios we analysed in this paper.

| Page 4: Deleted | Sagy Cohen | 3/28/16 11:40 AM |
|---|---|---|

.

| Page 4: Inserted | Sagy Cohen | 3/28/16 11:40 AM |
|---|---|---|

:

| Page 5: Inserted | Sagy Cohen | 3/22/16 4:58 PM |
|---|---|---|

 In Cohen et al. (2015) also outline and discuss the model assumptions.

| Page 5: Deleted | Sagy Cohen | 3/28/16 11:37 AM |
|---|---|---|

Below we describe the mARM5D equations that include the parameters that are modified by the simulation scenarios we analysed here.

| Page 5: Deleted | Sagy Cohen | 3/24/16 11:51 AM |
|---|---|---|

by

| Page 5: Inserted | Sagy Cohen | 3/24/16 11:51 AM |
|---|---|---|

using a modification of the TOPOG model (TOPOG, 1997) sediment transport equation

| Page 5: Deleted | Sagy Cohen | 3/24/16 11:52 AM |
|---|---|---|

a modified *Engelhund and Hansen* (1968) equation:

| Page 5: Inserted | Sagy Cohen | 3/22/16 5:30 PM |
|---|---|---|

$$q_s = e \frac{q^{n_1} S^{n_2}}{(s-1)^2 {d_{50}}^{n_3}}$$

| Page 5: Inserted | Sagy Cohen | 3/29/16 9:33 AM |
|---|---|---|

(m/m)

| Page 5: Inserted | Sagy Cohen | 3/28/16 2:44 PM |
|---|---|---|

$$\left[\frac{A}{A_p}\right]^{n_4} \frac{Q}{(A_p)^{0.5}}$$

| Page 6: Deleted | Sagy Cohen | 3/28/16 11:43 AM |
|---|---|---|

modeled

| Page 6: Inserted | Sagy Cohen | 3/28/16 11:43 AM |
|---|---|---|

modelled

| Page 6: Formatted | Sagy Cohen | 3/28/16 11:43 AM |
|---|---|---|

Not Highlight

| Page 6: Inserted | Sagy Cohen | 3/24/16 2:30 PM |
|---|---|---|

| Page 6: Deleted | Sagy Cohen | 3/28/16 11:13 AM |
|---|---|---|

$$D_s = S^\beta k \Delta t$$

| Page 6: Formatted | David | |
|---|---|---|

Lowered by  6 pt

| Page 6: Inserted | Sagy Cohen | 3/28/16 11:12 AM |
|---|---|---|

$$D_s = \left(\frac{S}{S_a}\right)^\beta D_o \Delta t$$

| Page 6: Inserted | Sagy Cohen | 3/28/16 11:13 AM |
|---|---|---|

| Page 6: Inserted | Sagy Cohen | 3/29/16 9:37 AM |
|---|---|---|

| Page 6: Deleted | Sagy Cohen | 3/29/16 10:29 AM |
|---|---|---|

| Page 6: Inserted | Sagy Cohen | 3/28/16 11:13 AM |
|---|---|---|

$D_o$

| Page 6: Deleted | Sagy Cohen | 3/28/16 11:13 AM |
|---|---|---|

$k$

| Page 6: Inserted | Sagy Cohen | 3/28/16 10:57 AM |
|---|---|---|

(m) and $S_a$ is the adjustment slope, the average slope in which $D_o$ was measured/estimated

| Page 6: Formatted | Sagy Cohen | 3/29/16 10:33 AM |
|---|---|---|

Font:Italic, Not Highlight

| Page 6: Formatted | Sagy Cohen | 3/29/16 10:33 AM |
|---|---|---|

Not Highlight

| Page 6: Inserted | Sagy Cohen | 3/29/16 10:30 AM |

Here we use $S_a$=0.2 which approximate our field site average slope.

| Page 6: Formatted | Sagy Cohen | 3/29/16 10:33 AM |

Not Highlight

| Page 6: Inserted | Sagy Cohen | 3/29/16 9:28 AM |

This value differs from the typical assumption of a linear relationship between slope and diffusion ($\beta$=1), suggesting that the influence of topographic slope in this soilscape is much lower. We will discuss this later.

| Page 6: Formatted | Sagy Cohen | 3/29/16 10:33 AM |

Not Highlight

| Page 6: Deleted | Sagy Cohen | 3/28/16 2:45 PM |

**2.1.3 Weathering**

The profile layers are subject to bedrock and soil weathering. We considered physical weathering calculated by breaking a parent particle into two daughter particles. As mass conservation is assumed, the diameters of the daughter particles ($d_1$, $d_2$) can be determined from the diameter of the parent particle ($d_0$). Based on experimental studies by *Wells et al.* (2008) we used a split-in-half geometry which leads to $d_1 = d_2$. A bedrock and soil weathering depth-dependency equation is used to set the weathering rate in each profile layer as a function of its depth below to the surface (*Heimsath et al.,* 1997). Following *Cohen et al.*, (2010), we used a modified version of the 'humped' soil-production function (*Ahert*, 1977) proposed by *Minasny and McBratney* (2006):

$$W_l = P_0 \left[ \exp(-\delta_1 h_1 + P_a) - \exp(-\delta_1 h_1) \right] \tag{5}$$

where $W_l$ is the physical weathering rate for profile layer $l$, $P_0$ and $P_a$ (mm/yr) is the potential (or maximum) and steady-state weathering rates respectively, $h_l$ (m) is the depth below the surface for layer $l$ and $\delta_1$ and $\delta_2$ are constants. The values proposed by *Minasny and McBratney* (2006), $\delta_1$=4, $\delta_2$=6, are used here.

| Page 6: Formatted | Sagy Cohen | 3/23/16 11:41 AM |

Highlight

| Page 6: Formatted | Sagy Cohen | 3/23/16 11:41 AM |

Highlight

| Page 6: Inserted | Sagy Cohen | 3/28/16 2:45 PM |

| Page 6: Inserted | Sagy Cohen | 3/28/16 2:44 PM |
|---|---|---|

$$h_s \underline{g_s}_{t+1}$$

| Page 6: Inserted | Sagy Cohen | 3/28/16 2:44 PM |
|---|---|---|

$$h_s \underline{g_s}_{t}$$

| Page 6: Inserted | Sagy Cohen | 3/28/16 2:44 PM |
|---|---|---|

$$K_a \underline{g}_{a}$$

| Page 6: Inserted | Sagy Cohen | 3/28/16 2:44 PM |
|---|---|---|

$$\underline{g_s}_{t}$$

[revised manuscript text omitted]

---

## Author Response (AR2)

**Author's Response to Reviewers' comments**

**Associate Editor**
The same two anonymous reviewers have read the resubmitted version of your manuscript, along with the response to reviewers that you provided. They reach differing conclusions (reject vs minor revisions). It is my opinion that many of the points that reviewer 1 raises, are fair. Indeed there appears to be insufficient consideration of the concerns that this reviewer raised based on the first version of the paper. More changes are clearly required. Reviewer 2, although less critical, agrees at least on one point: that one of your most important formulae has wrong units. To my mind, this means that adaptation is required, and that model reruns with reconsideration of results are necessary. I translate this into a request for resubmission after major revisions.

Should you decide to resubmit an improved manuscript, please make sure to provide a detailed response that reflects the importance of reviewer 1's concerns. We would like to thank the Editor, Associate Editor and two referees for their continued efforts in this process.

**Reviewer #1**
1) In my previous review I commented that the fact that a model with both fluvial and diffusive processes works better than one with just diffusive or just fluvial processes is not a significant conclusion. In their response the authors state that this is not a conclusion of their study (i.e., "the conclusion of this study is NOT that both transport mechanisms are in play..."). In fact, the text of the paper makes clear that this is major conclusion of their paper. Their conclusion section states "Only by simulating both fluvial and diffusive transport mechanisms can the model correctly simulate the observed soil distribution."
We did not intend to make this a major conclusion. However, it is an interesting insight that only when both processes are employed do we see the best outcome. There we see this as A conclusion not THE conclusion and it is stated in the context of the new insights of this study: *"Only when both fluvial and diffusive sediment transport mechanisms were modeled, a reasonable correspondence was achieved. While soilscapes are generally thought off as resulting from both transport mechanisms, this and previous studies demonstrates that fluvial-diffusion coupling is more pronounced in aeolian dominated soilscape."*

However, I accept the fact that the authors are also examining how the relative importance of diffusive versus fluvial processes varies with topographic position. Thank you.

However, the authors do not provide any reason why the addition of mass in the form of aeolian deposition would fundamentally alter the relative importance of fluvial versus diffusive processes as a function of topographic position, which was considered in Cohen et al. (2015) for what the authors call "bedrock (normal) landscapes".

We respectfully say that this statement is incorrect! Please see these two paragraphs in the introduction (and a whole separate paper on this topic – Cohen et al., 2015):

*"From a soilscape evolution point of view, aeolian dominated soilscapes differ from bedrock-weathering dominated soilscapes in several ways. In bedrock-weathering systems in situ weathering rates decrease exponentially with soil depth (Gilbert, 1877; Ahnert, 1977), thus regulating soil production as a function of regolith thickness (Heimsath et al., 1997). Weathering of regolith and soil leads to vertical particle size distribution with finer particles closer to the surface as a function of the soil and regolith age, namely time exposed to weathering (Yoo and Mudd 2008). At the surface, armouring can develop by size-selective entrainment (Kim and Ivanov, 2014) or vegetation shielding, which limits sediment transport by overland flow (Willgoose and Sharmeen, 2006). Given sufficient time and in the absence of vertical mixing due to pedoturbation, these processes - depth dependent weathering, vertical self-organization and surface armouring - will stabilize the soilscape leading to steady-state or dynamic equilibrium conditions (Cohen et al., 2013 & 2015). In aeolian dominated landscapes these controls on soil production and transport are largely ineffective as: (1) much of the soil is transported to the system as airborne sediments, i.e., no depth dependency; and (2) fine and highly erodible material is continuously deposited on top of older surface soils which limits the potential for surface armouring and vertical self-organization.*

*The differences between aeolian and bedrock-weathering dominated soilscapes lead us to conclude that traditional (i.e. bedrock weathering originated) soilscape evolution analysis is inappropriate for investigating the history of the aforementioned loess soilscapes. In Cohen et al. (2015) we developed a soilscape evolution model (mARM5D) to study the differences and interactions between aeolian and bedrock weathering soil production on a synthetic 1D hillslope. In that paper we have found that bedrock weathering dominated soilscapes are considerably more stable and showed much lower spatial (aerial) variability in soil depth and particle size distribution (PSD). We proposed that aeolian-dominated landscapes are more responsive to environmental changes (e.g., climatic and anthropogenic) compared with bedrock-weathering landscapes. "*

Instead, they simply modify the values of key parameters (n4 and beta especially) without calibration to data, then conclude that aeolian soilscapes lead to a completely different ratio of fluvial to diffusive processes as a function of topographic position than bedrock landscapes. I would be completely in support of this result if there was actual evidence presented that n4 and beta should be 0.1 for this field site or any other aeolian soilscape, but there is simply no data presented to demonstrate that these are the correct values. I continue to believe that any difference between the results of this paper and that of Cohen et al. (2015) for bedrock landscapes results not from the fact that one is a "bedrock (normal)" landscape and the other an "aeolian soilscape" but rather that this study has chosen very different and ad hoc values for n4 and beta that have no justification and no clear connection to this our any other aeolian-dominated landscape.

We respectfully disagree with the above comment. Cohen et al. (2015*) "The effects of sediment‐transport, **weathering and aeolian** mechanisms on soil evolution"* explored the difference between bedrock weathering and aeolian dominated soilscapes. In that paper we used n4 =1 and beta =1 as is commonly used. In this current manuscript we present a conceptual study of a specific case study – semi-arid, aeolian dominated, soil-depleted and high-gradient soilscape- focusing on differences between fluvial and diffusive transport mechanisms on this type of soilscape. We are NOT investigating the differences between aeolian and bedload dominated soilscapes in this paper! This comments most likely stem from the section in the discussion that focus on the differences between aeolian and bedrock soilscapes. We have revised it to clarify the point that the two studies cannot be directly compared.

Regarding the values of these two coefficients: we have found – using an extensive sensitivity analysis and discharge data for this site – that values of 0.1 provide the best approximation. We agree that this is not an ideal way of adjusting this variable and discuss that in the manuscript.  Resulting from this, we interpret our results in a conceptual framework and are very careful in stating anything which goes outside these limitations. We propose conceptual insights that could be tested with a more focused analysis.

 In their rebuttal the authors present two figures of soil diffusivity versus slope that purport to show a calibration, but there is no data in these figures. They appear to be simply graphs of S^1 and S^0.1.

Yes, we agree that they are and  we should have explained the plots better. The intention was to show the effect of beta on diffusion rate for a range of slope values. As can be seen, a b=0.1 (right plot) reduces the range of change down the hillslope. We suggest  that this is why a beta=0.1 is needed for this site given its steep concave-down shape (increasing slope gradient downslope). This assertion was added in the last manuscript revision and now developed even further  in the 2nd revision.

[Figure]

[Figure]

A more minor but related issue is that I don't think that bedrock (normal) landscapes and aeolian soilscapes are fundamentally different kinds of landforms. They exist on a continuum, since soils everywhere in the world contain some fraction of material derived from in situ weathering and some from aeolian deposition.

Yes, this is correct -but what about the extremes? There are many aeolian-dominated soilscapes which – as we described in the introduction and explored in our last paper – are potentially much different in some aspects. For example our analysis suggests that aeolian-dominated soilscapes are more susceptible to environmental changes.

This was my point when I recommended to the authors that they obtain some data (using immobile element ratios or similar geochemical techniques) on the relative fraction of the soil derived from aeolian input versus in situ weathering. The authors countered that they don't need data because the aeolian-dominated nature is clear from "walking the site." I don't know what this means.
We may have had a misinterpretation here. We said that this region was studied extensively for many years and it is in general agreement that the soil has a strong aeolian input. The relevant literature is cited in the manuscript.

2) I agree that some type of power-law relationship with depth and slope is correct for fluvial transport, so I am not going to harp on the fact that there is little clarity in how the authors have modified Engelund and Hansen (1967) to arrive at their eqn. (1). The fact remains that, if the reader looks at p. 42 of Engelund and Hansen (actually p. 41) there is an expression for the Shields stress as a function of the mean dune height and other parameters. How these equations relates to equation (1) of Cohen et al. remains unclear. Referring me to TOPOG or some other model does not answer the question of how one goes, step by step, from one or more of Engelund and Hansen's equations to eqn. (1) of Cohen et al., which is what I think readers need (in an appendix would be fine, if the authors consider this to be ancillary).
As we acknowledged in our response that directly referencing Eq 1 to Engelund and Hansen (1967) was not the best approach even though it is the equation origin. The equation was adopted from the TOPOG model and that was clearly referenced in the last revision. We have now also added a reference to Merrit et al (2003) review paper where they outline this link. TOPOG is a well used and vetted model and we don't think we need to retrace their adaptation step-by-step.

3) Yair and Kossovsky (2012) may provide a basis for using a value of n4 that is lower than 1. However, the authors have provided no justification for the specific value they used (0.1) either in the paper or in their rebuttal. This value still seems to be pulled from thin air. Why not 0.3 or 0.5? This is a major issue, since the value of n4 directly controls the strength of the fluvial term in the model, and its variation with topographic position.
See response to points 1 and 2 above.

4) I am very disturbed by the author's statement that "Diffusion absolutely involved routing, how else does the model transport the sediment down the slope?" This seems to indicate that the authors do not know how to model diffusion. Diffusion is the divergence of a flux. The divergence of a flux is defined as the derivative of the x component of the flux in the x direction plus the derivative of the y component of the flux in the y direction. Flow routing such as D8 do not appear anywhere in the

definition of divergence and SHOULD NOT be used for diffusion on hillslopes. There is certainly no reason why flow routing methods are REQUIRED to model diffusion. The fact that D8 is being incorrectly used leads directly to the unrealistic "striping" seen in the results. Obviously, any model result that shows large variations in model results along 45 or 90 degrees is a model artifact that needs to be minimized. I don't see any of this kind of striping in other aeolian soilscape models that have been used in the literature to model soil depth or aeolian soil fraction (which are not referenced). This is a major issue that simply must be fixed.

The reviewer is incorrect. The flux form for diffusion is qs=DS=-D dz/dx. This is fundamentally how diffusion is defined based on a gradient of the constituent being diffused. The 2nd order finite difference term commonly used is subsequently defined based on mass continuity equations (the divergence term mentioned by the reviewer) using this flux equation so that dz/dt=-D (dz2/dx2+dz2/dy2) and the finite difference approximation typically discussed is derived directly from this. Mathematically these two forms for diffusion are equivalent. The approximations that is used in mARM5D uses the flux form and follows the approach used in the first of the landform evolution models SIBERIA (Willgoose et al 1991). Essentially the D8 is used to determine the flux direction, and the amount of flux per time step is determined by the flux form of the diffusion equation. SIBERIA originally used the finite difference form of the equation but subsequent testing showed the flux form gave very similar results while being MUCH easier to implement for (1) irregular boundaries and (2) variable spaced grids. The reviewer is right in one regard … doing flow routing is not REQUIRED … its just easier when used in conjunction with other processes (e.g. erosion) and more realistic problems with irregular boundaries and variably spaced grids or TINS.

5) In my review I noted that D8 cannot be used for hillslopes, where the flow of water and fluvial sediment is divergent (D8 assumes strong convergence everywhere). The authors counter that they have developed a fast model and so any inaccuracies should be acceptable in the name of speed. I disagree. I think the model needs to be accurate first. Again, this is a major issue that must be addressed.

Each drainage direction algorithm has its strengths and weaknesses. To the best of our knowledge most models use D8 routing (some even use D4) as it is a much more efficient algorithm. Yes, we could start to employ different routing algorithms but this would greatly complicate an already complex modelling study. We see this as an important next step. We also recognize that efficiency is important to these kind of studies as we may not have even go this far if we started working through drainage routing options. We argue that the while a more flexible algorithm may help resolve some of the problems we observed in our simulations (and fully discussed) the overall insight from this study would be the same. We acknowledged that this is a needed improvement in the model.

6) As with the value of n4, no calibration is performed to show that beta = 0.1 for this field site, either in the paper or in the rebuttal. To do this calibration, the authors would need data. There is no data in the two figures provided in the rebuttal, which appear to simply be plots of S^1 and S^0.1. The authors may be

correct that beta = 0.1 is an interesting result, but the authors need to show the readers (via some type of least-squares or other fit to DATA) why beta = 0.1 at this location.

See response to comments 1 and 2.

7) Equation (4) continues to have a units problem. In their rebuttal the authors state that equation (4) does not have a time step but I am looking at eqn. (4) right now and the last term is "delta t".

You are correct – we thought you were referring to equation 3.

Since diffusivity is always $L^2/T$ and the only other term with units is delta t then D must have units of $L^2$. However, the reported units are $L/T$. The authors may think that "In practice it makes not difference" but I think it will matter to readers trying to replicate their work. The authors have indicated that their diffusivity has units of L (no units were reported in the paper so I was guessing that diffusivity had units of $L^2/T$, which is always the proper unit of diffusivity), but this still leads to a units problem since if I use units of L for diffusivity, then according to eqn. (4) D should have units of $L*T$, not $L/T$.

The model calculates the **proportion** of each layer that is displaced due to diffusion. That proportion decrease exponentially down the soil profile (Eq. 3) in proportion to the surface diffusion rate (eq. 4). In effect the PSD vector for each layer in each grid-cell is deducted by the multiplication of the PSD vector by the proportion of the layer that has been displaced (calculated by dividing the surface diffusion by cell size) plus a temporal adjustment parameter. We have simplified the description of this algorithm as it can be confusing without understanding the model unique architecture (which is shortly describe at the start of the section with references to previous papers). We have now revised the diffusion description sub-section.

8) I am not going to comment on the author's response to my concerns about the weathering portion of their model, since I don't find it acceptable for the authors to refer me to older papers and send me code rather than simply answering my question.

We apologise as there is some misunderstanding here. We had presented over a page of response just for the 1st reviewer's comments (see box below) on this issue and consider that we have a satisfactory response

*When I tracked down Minasny and McBratney (2006) I found a rather different equation (their equation (4)). Equation (5) is dimensionally incorrect. It is wrong to have the steady state weathering rate appear inside the exponential – the argument of any exponential should be unitless.* The equation was indeed (as stated) modified, the Po parameter was moved to allow for an above-zero watering rate at the surface while depth rates down the soil profile asymptote to zero. Cohen et al. (2010) focused on the model weathering equations and algorithm. We can see how the description of the model weathering calculations is confusing and misrepresentative. This stems, again, from our attempt to simplify a complex algorithm into one equation with time-varying parameter. The full algorithm includes compiling a transition matrix which control the

transition of PSD in each particle size class to smaller class(s) in each soil-profile layer. The algorithm was described at length in Cohen et al. (2009 and 2010). The depth-varying weathering rate equation in the model (see the actual model FORTRAN function below) is not directly used to calculate weathering rate, rather the relative (normalized) change in weathering down the profile. As part of the revised model description we removed the section describing the weathering calculations as it does not include parameters that are modified in the simulation scenarios we analyzed in this paper.

```
!####################################################################
! A function to calculate the decline in weathering rate as a function of depth
! It return the WeatheringAlpha which is different for every layer
! The exponential function is taken from Minasny & McBratney (2006)
! de/dt=Po{Exp(-k1h)-Exp(-k2h)}+Pa ;
! Their original values are: Po-potential WR=0.25; k1=4; k2=6; Pa- steady state WR=0.005
! The function was changed in version 4.5.1 to account to close to zero weathering in
lower layers
!####################################################################
    REAL*8 Function DepthWeatheringAlpha(WeatherAlpha,i,LayerDepth)
    IMPLICIT NONE
    Integer i
    Real*8 WeatherAlpha, LayerDepth, Ratio, Depth
    If (i==1) Then
    Depth=0.5
    Else
    Depth=(i-1)*LayerDepth-(LayerDepth/2)
    End If
    Depth=Depth/100 !convert to meters
    Ratio=(0.25*((EXP(-4*Depth+0.02))-(EXP(-6*Depth))+0))/0.04 !We divide by 0.04 to
normalize it
    DepthWeatheringAlpha = WeatherAlpha*Ratio
    Return
    End Function DepthWeatheringAlpha
```

*Why are delta_1 and delta_2 equal to 4 and 6? What are the units? If they are meters these are very large values (i.e. they imply that weathering rates fall off by a factor of e only once the soil is at least 4 m thick. This is a very thick soil).* Following on the comment above, the model algorithm uses the Minasny and McBratney (2006) equation to get the relative change in weathering rate down the soil profile. These variables therefor control the shape rather then the actual weathering rate. Their values were based on Minasny and McBratney (2006) to maintain the relative change (i.e. shape of the hump function) and so in reality they are unitless.

9) I don't see how "walking the site" allows one to conclude that the fine grained component of the soil cannot be from weathering of bedrock. Again, data is needed. Aeolian deposition rates and soil formations were well studied in this region. References are provided in the manuscript and the aeolian rate was based on these. There is simply no need to go into further geochemical complexity and we consider the request to be unreasonable. The bedrock is limestone which is well known for low soil production rates – we made a simple assumption (which is clearly stated in

the manuscript) that weathering rate is an order of magnitude lower than aeolian deposition (at baseline levels).

10) I asked for a table of parameters. The authors refer me to the last 4 papers on this model. Including a table is a very simple request and I am disappointed that the authors refuse to do even that much to assist the reader.
The key parameters (with their symbology) are outlined in the methodology and again in Table 1.

I can see how the authors might view my review as an attack on their paper. However, I am really trying to help them meet common standards for accuracy and transparency.
We very much appreciate the comments. Thanks!

**Reviewer #2**
The authors have addressed all of the reviewers' comments. However the humped equation should be corrected, so that the units are correct:
The equation as presented in Minasny & Mcratney (2006)
$de/dt = P0 * (exp(-k1*h) - exp(-k2*h) + Pa$
where P0 is the potential weathering rate (m/y), Pa weathering rate at steady-state and k1 and and k2 are rate constants (units 1/m).

The current formulation does not yield a correct unit:
$Wl = P0 \exp(-\delta 1h1 + Pa) - \exp(-\delta 1h1)]$
As we stated in the last response – we modified their equation to a unitless ratio. Here is our response to reviewer #1 in the last review:
*"The equation was indeed (as stated) modified, the Po parameter was moved to allow for an above-zero watering rate at the surface while depth rates down the soil profile asymptote to zero. Cohen et al. (2010) focused on the model weathering equations and algorithm. We can see how the description of the model weathering calculations is confusing and misrepresentative. This stems, again, from our attempt to simplify a complex algorithm into one equation with time-varying parameter. The full algorithm includes compiling a transition matrix which control the transition of PSD in each particle size class to smaller class(s) in each soil-profile layer. The algorithm was described at length in Cohen et al. (2009 and 2010). The depth-varying weathering rate equation in the model (see the actual model FORTRAN function below) is not directly used to calculate weathering rate, rather the relative (normalized) change in weathering down the profile. As part of the revised model description we removed the section describing the weathering calculations as it does not include parameters that are modified in the simulation scenarios we analyzed in this paper."*

**Main document changes and comments**

| Page 5: Inserted | Sagy Cohen | 5/25/16 3:50 PM |
|---|---|---|

; Merritt et al., 2003

| Page 5: Deleted | Sagy Cohen | 5/25/16 3:48 PM |
|---|---|---|

$$q_s = e \frac{q^{n_1} S^{n_2}}{(1-s)^2 d_{50}{}^{n_3}} \Delta t$$

| Page 5: Formatted | red carrot |
|---|---|

Lowered by 16 pt

| Page 5: Inserted | Sagy Cohen | 5/25/16 3:49 PM |
|---|---|---|

$$\Delta t$$

| Page 5: Formatted | Sagy Cohen | 5/26/16 10:45 AM |
|---|---|---|

Font:(Default) +Theme Body

| Page 5: Inserted | Sagy Cohen | 5/25/16 3:48 PM |
|---|---|---|

| Page 5: Inserted | Sagy Cohen | 5/25/16 3:49 PM |
|---|---|---|

| Page 5: Inserted | Sagy Cohen | 5/27/16 1:59 PM |
|---|---|---|

Using an extensive parametric study (not presented here) we have found that $n_4$=0.1 leads to best approximation of observed soil distribution. We will discuss this later.

| Page 5: Formatted | Sagy Cohen | 6/17/16 2:39 PM |
|---|---|---|

Not Highlight

| Page 5: Deleted | Sagy Cohen | 5/27/16 1:59 PM |
|---|---|---|

We therefore use $n_4$=0.1 in this study.

| Page 5: Inserted | Sagy Cohen | 6/17/16 2:38 PM |
|---|---|---|

.

| Page 6: Deleted | Sagy Cohen | 5/27/16 3:35 PM |
|---|---|---|

$$D_l = D_s [\exp(-\lambda h_l)]$$

| Page 6: Formatted | red carrot |
|---|---|

Lowered by 6 pt

| Page 6: Inserted | Sagy Cohen | 5/27/16 3:34 PM |
|---|---|---|

$$Dc_l =$$

| Page 6: Deleted | Sagy Cohen | 5/27/16 3:38 PM |
|---|---|---|

| Page 6: Inserted | Sagy Cohen | 5/27/16 3:34 PM |

$$(-\lambda h_l)$$

| Page 6: Inserted | Sagy Cohen | 5/27/16 3:35 PM |

$c$

| Page 6: Deleted | Sagy Cohen | 5/27/16 3:37 PM |

(m/y)

| Page 6: Inserted | Sagy Cohen | 5/27/16 3:36 PM |

fraction of diffusion rate

| Page 6: Deleted | Sagy Cohen | 5/27/16 3:36 PM |

diffusive transport rate

| Page 6: Inserted | Sagy Cohen | 5/27/16 3:36 PM |

relative to the diffusion rate at the surface layer ($l_s$)

| Page 6: Formatted | Sagy Cohen | 5/27/16 3:36 PM |

Font:Not Italic

| Page 6: Formatted | Sagy Cohen | 5/27/16 3:37 PM |

Font:Italic

| Page 6: Formatted | Sagy Cohen | 5/27/16 3:37 PM |

Font:Italic, Subscript

| Page 6: Deleted | Sagy Cohen | 5/27/16 3:39 PM |

$D_s$ is the surface (maximum) diffusive sediment transport rate (m/y),

| Page 6: Inserted | Sagy Cohen | 5/27/16 3:40 PM |

/y

[revised manuscript text omitted]

---

## Author Response (AR3)

**Author's Response to Reviewers' comments**

**Associate Editor**
The same two anonymous reviewers have read the resubmitted version of your manuscript, along with the response to reviewers that you provided. They reach differing conclusions (reject vs minor revisions). It is my opinion that many of the points that reviewer 1 raises, are fair. Indeed there appears to be insufficient consideration of the concerns that this reviewer raised based on the first version of the paper. More changes are clearly required. Reviewer 2, although less critical, agrees at least on one point: that one of your most important formulae has wrong units. To my mind, this means that adaptation is required, and that model reruns with reconsideration of results are necessary. I translate this into a request for resubmission after major revisions.

Should you decide to resubmit an improved manuscript, please make sure to provide a detailed response that reflects the importance of reviewer 1's concerns. We would like to thank the Editor, Associate Editor and two referees for their continued efforts in this process.

**Reviewer #1**
1) In my previous review I commented that the fact that a model with both fluvial and diffusive processes works better than one with just diffusive or just fluvial processes is not a significant conclusion. In their response the authors state that this is not a conclusion of their study (i.e., "the conclusion of this study is NOT that both transport mechanisms are in play…"). In fact, the text of the paper makes clear that this is major conclusion of their paper. Their conclusion section states "Only by simulating both fluvial and diffusive transport mechanisms can the model correctly simulate the observed soil distribution."
We did not intend to make this a major conclusion. However, it is an interesting demonstration of how both processes effect soilscape evolution under these unique conditions. Hancock et al (2002) demonstrated that the inclusion of diffusion was extremely important for correctly simulating landscape evolution. We see this statement as part of the conclusion not THE conclusion and it is stated in the context of the new insights of this study: *"Only when both fluvial and diffusive sediment transport mechanisms were modeled, a reasonable correspondence was achieved. While soilscapes are generally thought off as resulting from both transport mechanisms, this and previous studies demonstrates that fluvial-diffusion coupling is more pronounced in aeolian dominated soilscape."*

However, I accept the fact that the authors are also examining how the relative importance of diffusive versus fluvial processes varies with topographic position. Thank you.

However, the authors do not provide any reason why the addition of mass in the form of aeolian deposition would fundamentally alter the relative importance of

fluvial versus diffusive processes as a function of topographic position, which was considered in Cohen et al. (2015) for what the authors call "bedrock (normal) landscapes".

We respectfully disagree. Please see these two paragraphs in the introduction (and a whole separate paper on this topic – Cohen et al., 2015):

*"From a soilscape evolution point of view, aeolian dominated soilscapes differ from bedrock-weathering dominated soilscapes in several ways. In bedrock-weathering systems in situ weathering rates decrease exponentially with soil depth (Gilbert, 1877; Ahnert, 1977), thus regulating soil production as a function of regolith thickness (Heimsath et al., 1997). Weathering of regolith and soil leads to vertical particle size distribution with finer particles closer to the surface as a function of the soil and regolith age, namely time exposed to weathering (Yoo and Mudd 2008). At the surface, armouring can develop by size-selective entrainment (Kim and Ivanov, 2014) or vegetation shielding, which limits sediment transport by overland flow (Willgoose and Sharmeen, 2006). Given sufficient time and in the absence of vertical mixing due to pedoturbation, these processes - depth dependent weathering, vertical self-organization and surface armouring - will stabilize the soilscape leading to steady-state or dynamic equilibrium conditions (Cohen et al., 2013 & 2015). In aeolian dominated landscapes these controls on soil production and transport are largely ineffective as: (1) much of the soil is transported to the system as airborne sediments, i.e., no depth dependency; and (2) fine and highly erodible material is continuously deposited on top of older surface soils which limits the potential for surface armouring and vertical self-organization.*

*The differences between aeolian and bedrock-weathering dominated soilscapes lead us to conclude that traditional (i.e. bedrock weathering originated) soilscape evolution analysis is inappropriate for investigating the history of the aforementioned loess soilscapes. In Cohen et al. (2015) we developed a soilscape evolution model (mARM5D) to study the differences and interactions between aeolian and bedrock weathering soil production on a synthetic 1D hillslope. In that paper we have found that bedrock weathering dominated soilscapes are considerably more stable and showed much lower spatial (aerial) variability in soil depth and particle size distribution (PSD). We proposed that aeolian-dominated landscapes are more responsive to environmental changes (e.g., climatic and anthropogenic) compared with bedrock-weathering landscapes. "*

Instead, they simply modify the values of key parameters (n4 and beta especially) without calibration to data, then conclude that aeolian soilscapes lead to a completely different ratio of fluvial to diffusive processes as a function of topographic position than bedrock landscapes. I would be completely in support of this result if there was actual evidence presented that n4 and beta should be 0.1 for this field site or any other aeolian soilscape, but there is simply no data presented to demonstrate that these are the correct values. I continue to believe that any difference between the results of this paper and that of Cohen et al. (2015) for bedrock landscapes results not from the fact that one is a "bedrock (normal)" landscape and the other an "aeolian soilscape" but rather that this study has chosen

very different and ad hoc values for n4 and beta that have no justification and no clear connection to this our any other aeolian-dominated landscape.

We respectfully disagree with the above comment. Cohen et al. (2015*) "The effects of sediment‐transport, **weathering and aeolian** mechanisms on soil evolution"* explored the difference between bedrock weathering and aeolian dominated soilscapes. In that paper we used n4 =1 and beta =1 as is commonly used. In this current manuscript we present a conceptual study of a specific case study – semi-arid, aeolian dominated, soil-depleted and high-gradient soilscape- focusing on differences between fluvial and diffusive transport mechanisms on this type of soilscape. We are NOT investigating the differences between aeolian and bedload dominated soilscapes in this paper! This comments most likely stem from the section in the discussion that focus on the differences between aeolian and bedrock soilscapes. We have revised it to clarify the point that the two studies cannot be directly compared.

Regarding the values of these two coefficients- in response to the reviewer and associate editor comments, we have conducted a new analysis for a 1D hillslope in our field site (near transects we discuss in the manuscript). To isolate the effects of n4 and beta values on the model results we ran these 1D simulations without the temporal change scenario. Simulating without the change scenario change the overall transport rates and the soilscape evolution and is therefore not readily comparable to the landscape simulations presented in the manuscript. This allows us, however, to adjust the transport rating (due to differences in the beta and n4 coefficients) to more clearly isolate the effects of beta and n4. We compared the following simulations:

n4 and beta = 1
n4 = 0.1 and beta = 1
n4 = 1 and beta = 0.1
n4 = 0.1 and beta = 0.1

Since the change in beta and n4 effect the fluvial and diffusive transport rates (=0.1 will reduce the rate), we first had to adjust these rates so the resulting soil distribution will be, as much as possible, due to differences in coefficient value and not due to differences in transport rate. We did that by finding (by calibration) the rate adjustment (increase) factor resulting in total soil depth that is similar to a simulation in which both coefficients equal 1.  We found that a factor of 4 is needed for when n4=0.1 and a factor on 1.2 is needed for when beta=0.1.

The results show that changing only one of the coefficients (n4 or beta) to 0.1 resulted in relatively little difference in soil distribution down this concave hillslope:

[Figure]

When both n4 and beta equal 0.1 the soil distribution was very similar at the upslope section of the hillslope but much deeper soil at the downslope section:

[Figure]

This soil distribution shape is quite common in our field site, a hump of deep soil driven by the balance between slope and area values (high slope mainly increase diffusion while high area increase fluvial transport). It could be argued that similar distribution could be achieved by adjusting the transport rates with n4 and beta equal to 1. Our analysis showed that this was not the case given the high concave down slopes of our field site.

This analysis is further expended in a separate study which is forthcoming. It is too complex and may be incomplete to expand in this current manuscript.

In their rebuttal the authors present two figures of soil diffusivity versus slope that purport to show a calibration, but there is no data in these figures. They appear to be simply graphs of S^1 and S^0.1.
Yes, we agree that they are and we should have explained that plots better. The intention was to show the effect of beta on diffusion rate for a range of slope values. As can be seen, a b=0.1 (right plot) reduces the range of change down the hillslope.

We suggest that this is why a beta=0.1 is needed for this site given its steep concave-down shape (increasing slope gradient downslope). This assertion was added in the last manuscript revision and now developed even further in the 2nd revision.

[Figure]

[Figure]

A more minor but related issue is that I don't think that bedrock (normal) landscapes and aeolian soilscapes are fundamentally different kinds of landforms. They exist on a continuum, since soils everywhere in the world contain some fraction of material derived from in situ weathering and some from aeolian deposition.

Yes, this is correct -but what about the extremes where aeolian deposition has considerable influence on soil depth and properties – that is the extreme end of the soilscape spectrum? There are many aeolian-dominated soilscapes which – as we described in the introduction and explored in our last paper – are potentially much different in some aspects. For example our analysis suggests that aeolian-dominated soilscapes are more susceptible to environmental changes. Ultimately, that is really what we are exploring here.

This was my point when I recommended to the authors that they obtain some data (using immobile element ratios or similar geochemical techniques) on the relative fraction of the soil derived from aeolian input versus in situ weathering. The authors countered that they don't need data because the aeolian-dominated nature is clear from "walking the site." I don't know what this means.

We may have had a misinterpretation here. We said that this region was studied extensively for many years and it is well established that the soil has a strong aeolian input. The relevant literature is cited in the manuscript.

2) I agree that some type of power-law relationship with depth and slope is correct for fluvial transport, so I am not going to harp on the fact that there is little clarity in how the authors have modified Engelund and Hansen (1967) to arrive at their eqn. (1). The fact remains that, if the reader looks at p. 42 of Engelund and Hansen (actually p. 41) there is an expression for the Shields stress as a function of the mean dune height and other parameters. How these equations relates to equation (1) of Cohen et al. remains unclear. Referring me to TOPOG or some other model does not answer the question of how one goes, step by step, from one or more of Engelund and Hansen's equations to eqn. (1) of Cohen et al., which is what I think readers need (in an appendix would be fine, if the authors consider this to be ancillary).

As we acknowledged in our response that directly referencing Eq 1 to Engelund and Hansen (1967) was not the best approach even though it is the equation origin. The equation was adopted from the TOPOG model and that was clearly referenced in the last revision. We have now also added a reference to Merrit et al (2003) review paper where they outline this link. TOPOG is a well used and vetted model and we don't think we need to retrace their adaptation step-by-step.

3) Yair and Kossovsky (2012) may provide a basis for using a value of n4 that is lower than 1. However, the authors have provided no justification for the specific value they used (0.1) either in the paper or in their rebuttal. This value still seems to be pulled from thin air. Why not 0.3 or 0.5? This is a major issue, since the value of n4 directly controls the strength of the fluvial term in the model, and its variation with topographic position.
We tested a range of values and found 0.1 to better match observed soil distribution. We agree that this is an intriguing result that merit further investigation. This is now discussed in the manuscript.

4) I am very disturbed by the author's statement that "Diffusion absolutely involved routing, how else does the model transport the sediment down the slope?" This seems to indicate that the authors do not know how to model diffusion. Diffusion is the divergence of a flux. The divergence of a flux is defined as the derivative of the x component of the flux in the x direction plus the derivative of the y component of the flux in the y direction. Flow routing such as D8 do not appear anywhere in the definition of divergence and SHOULD NOT be used for diffusion on hillslopes. There is certainly no reason why flow routing methods are REQUIRED to model diffusion. The fact that D8 is being incorrectly used leads directly to the unrealistic "striping" seen in the results. Obviously, any model result that shows large variations in model results along 45 or 90 degrees is a model artifact that needs to be minimized. I don't see any of this kind of striping in other aeolian soilscape models that have been used in the literature to model soil depth or aeolian soil fraction (which are not referenced). This is a major issue that simply must be fixed.
We have extended the description of these features and now provide a more concise explanation.
Regarding the use of D8 for diffusion transport - the flux form for diffusion is qs=DS=-D dz/dx. This is fundamentally how diffusion is defined based on a gradient of the constituent being diffused. The 2nd order finite difference term commonly used is subsequently defined based on mass continuity equations (the divergence term mentioned by the reviewer) using this flux equation so that dz/dt=-D (dz2/dx2+dz2/dy2) and the finite difference approximation typically discussed is derived directly from this. Mathematically these two forms for diffusion are equivalent. The approximations that is used in mARM5D uses the flux form and follows the approach used in the first of the landform evolution models SIBERIA (Willgoose et al 1991). Essentially the D8 is used to determine the flux direction, and the amount of flux per time step is determined by the flux form of the diffusion equation. SIBERIA originally used the finite difference form of the equation but subsequent testing showed the flux form gave very similar results while being

MUCH easier to implement for (1) irregular boundaries and (2) variable spaced grids.

However, the reviewer is right … doing flow routing is not required … it is included and used in conjunction with other processes (e.g. erosion) and more realistic problems with irregular boundaries and variably spaced grids or TINS.

5) In my review I noted that D8 cannot be used for hillslopes, where the flow of water and fluvial sediment is divergent (D8 assumes strong convergence everywhere). The authors counter that they have developed a fast model and so any inaccuracies should be acceptable in the name of speed. I disagree. I think the model needs to be accurate first. Again, this is a major issue that must be addressed.
Yes, we agree. This was poorly worded. Each drainage direction algorithm has its strengths and weaknesses. To the best of our knowledge most models use D8 routing (some even use D4, i.e. CAESAR) as it is a much more efficient algorithm. Yes, we could start to employ different routing algorithms but this would greatly complicate an already complex modeling study. In fact we do not know of any model that differentiates between drainage direction based on convergence/divergence. However we see this as an important next step and added this comment to the manuscript. We also recognize that efficiency is important to these kind of studies. We argue that the while a more flexible algorithm may help resolve some of the problems we observed in our simulations (which are discussed in the manuscript) the overall insight from this study would be the same. We acknowledged that this is a needed improvement in the model (and other models).

6) As with the value of n4, no calibration is performed to show that beta = 0.1 for this field site, either in the paper or in the rebuttal. To do this calibration, the authors would need data. There is no data in the two figures provided in the rebuttal, which appear to simply be plots of S^1 and S^0.1. The authors may be correct that beta = 0.1 is an interesting result, but the authors need to show the readers (via some type of least-squares or other fit to DATA) why beta = 0.1 at this location.
See response to comments 1 and 2.

7) Equation (4) continues to have a units problem. In their rebuttal the authors state that equation (4) does not have a time step but I am looking at eqn. (4) right now and the last term is "delta t".
You are correct – we thought you were referring to equation 3.

Since diffusivity is always L^2/T and the only other term with units is delta t then D must have units of L^2. However, the reported units are L/T. The authors may think that "In practice it makes not difference" but I think it will matter to readers trying to replicate their work. The authors have indicated that their diffusivity has units of L (no units were reported in the paper so I was guessing that diffusivity had units of L^2/T, which is always the proper unit of diffusivity), but this still leads to a units

problem since if I use units of L for diffusivity, then according to eqn. (4) D should have units of L*T, not L/T.

The model calculates the **proportion** of each layer that is displaced due to diffusion (L relative to L of the cell). That proportion decrease exponentially down the soil profile (Eq. 3) in proportion to the surface diffusion rate (eq. 4). In effect the PSD vector for each layer in each grid-cell is deducted by the multiplication of the PSD vector by the proportion of the layer that has been displaced (calculated by dividing the surface diffusion by cell size) plus a temporal adjustment parameter. We have simplified the description of this algorithm as it can be confusing without understanding the model's unique architecture (which is shortly describe at the start of the section with references to previous papers). We have revised the diffusion description sub-section to clarify this point. $D_s$ is in units of m by multiplying $D_o$ (m/y) by dt.  The actual change to soil is now showen in eq 5. Ds is divided by a grid cell length (m) to calculate a unitless proportion of sediment transported from a grid cell.

8) I am not going to comment on the author's response to my concerns about the weathering portion of their model, since I don't find it acceptable for the authors to refer me to older papers and send me code rather than simply answering my question.

We apologize, as there is some misunderstanding here. We had presented over a page of response just for the 1st reviewer's comments (see box below) on this issue:

*When I tracked down Minasny and McBratney (2006) I found a rather different equation (their equation (4)). Equation (5) is dimensionally incorrect. It is wrong to have the steady state weathering rate appear inside the exponential – the argument of any exponential should be unitless.* The equation was indeed (as stated) modified, the Po parameter was moved to allow for an above-zero watering rate at the surface while depth rates down the soil profile asymptote to zero. Cohen et al. (2010) focused on the model weathering equations and algorithm. We can see how the description of the model weathering calculations is confusing and misrepresentative. This stems, again, from our attempt to simplify a complex algorithm into one equation with time-varying parameter. The full algorithm includes compiling a transition matrix which control the transition of PSD in each particle size class to smaller class(s) in each soil-profile layer. The algorithm was described at length in Cohen et al. (2009 and 2010). The depth-varying weathering rate equation in the model (see the actual model FORTRAN function below) is not directly used to calculate weathering rate, rather the relative (normalized) change in weathering down the profile. As part of the revised model description we removed the section describing the weathering calculations as it does not include parameters that are modified in the simulation scenarios we analyzed in this paper.

```
!####################################################################
! A function to calculate the decline in weathering rate as a function of depth
! It return the WeatheringAlpha which is different for every layer
```

```
! The exponential function is taken from Minasny & McBratney (2006)
! de/dt=Po{Exp(-k1h)-Exp(-k2h)}+Pa ;
! Their original values are: Po-potential WR=0.25; k1=4; k2=6; Pa- steady state WR=0.005
! The function was changed in version 4.5.1 to account to close to zero weathering in
lower layers
!#####################################################################
    REAL*8 Function DepthWeatheringAlpha(WeatherAlpha,i,LayerDepth)
    IMPLICIT NONE
    Integer i
    Real*8 WeatherAlpha, LayerDepth, Ratio, Depth
    If (i==1) Then
    Depth=0.5
    Else
    Depth=(i-1)*LayerDepth-(LayerDepth/2)
    End If
    Depth=Depth/100 !convert to meters
    Ratio=(0.25*((EXP(-4*Depth+0.02))-(EXP(-6*Depth))+0))/0.04 !We divide by 0.04 to
normalize it
    DepthWeatheringAlpha = WeatherAlpha*Ratio
    Return
    End Function DepthWeatheringAlpha
```

*Why are delta_1 and delta_2 equal to 4 and 6? What are the units? If they are meters these are very large values (i.e. they imply that weathering rates fall off by a factor of e only once the soil is at least 4 m thick. This is a very thick soil).* Following on the comment above, the model algorithm uses the Minasny and McBratney (2006) equation to get the relative change in weathering rate down the soil profile. These variables therefor control the shape rather then the actual weathering rate. Their values were based on Minasny and McBratney (2006) to maintain the relative change (i.e. shape of the hump function) and so in reality they are unitless.

9) I don't see how "walking the site" allows one to conclude that the fine grained component of the soil cannot be from weathering of bedrock. Again, data is needed. Yes, this is a poor choice of words. Aeolian deposition rates and soil formations were well studied in this region. References are provided in the manuscript and the aeolian rate was based on these. There is simply no need to go into further geochemical complexity and we consider the request to be unreasonable. The bedrock is limestone which is well known for low soil production rates – we made a simple assumption (which is clearly stated in the manuscript) that weathering rate is an order of magnitude lower than aeolian deposition (at baseline levels).

10) I asked for a table of parameters. The authors refer me to the last 4 papers on this model. Including a table is a very simple request and I am disappointed that the authors refuse to do even that much to assist the reader.
The key parameters (with their symbology) are outlined in the methodology and again in Table 1.

I can see how the authors might view my review as an attack on their paper. However, I am really trying to help them meet common standards for accuracy and transparency.

We very much appreciate the comments. Thanks!

**Reviewer #2**
The authors have addressed all of the reviewers' comments. However the humped equation should be corrected, so that the units are correct:
The equation as presented in Minasny & Mcratney (2006)
de/dt = P0 * (exp(-k1*h) - exp(-k2*h) + Pa
where P0 is the potential weathering rate (m/y), Pa weathering rate at steady-state and k1 and and k2 are rate constants (units 1/m).

The current formulation does not yield a correct unit:
Wl = P0 exp(−δ1h1 + Pa )− exp(−δ1h1)]
The equation was modified into a normalized (unitless) change in weathering rate relative to surface weathering rate. The Po parameter was moved to allow for an above-zero watering rate at the surface while depth rates down the soil profile asymptote to zero. Cohen et al. (2010) focused on the model weathering equations and algorithm. We can see how the description of the model weathering calculations is confusing and misrepresentative out of the context of the full model algorithm. This stems, again, from our attempt to simplify a complex algorithm into one equation with time-varying parameter. The full algorithm includes compiling a transition matrix which control the transition of PSD in each particle size class to smaller class(s) in each soil-profile layer. The algorithm was described at length in Cohen et al. (2009 and 2010). The depth-varying weathering rate equation in the model is not directly used to calculate weathering rate, rather the relative (normalized) change in weathering down the profile. As part of the revised model description we removed the section describing the weathering calculations as it does not include 
[revised manuscript text omitted]

Sagy Cohen 11/21/2016 11:20 AM

Sagy Cohen 11/21/2016 11:20 AM

Sagy Cohen 11/21/2016 11:20 AM

Sagy Cohen 11/21/2016 11:20 AM

Sagy Cohen 11/21/2016 11:20 AM

Sagy Cohen 11/21/2016 11:20 AM

Sagy Cohen 11/21/2016 11:20 AM

**2.1.2 Diffusion transport**

Traditionally, equations of two-dimensional diffusive transport calculate sediment discharge as a linear relationship to slope, soil thickness and a diffusion coefficient (e.g. the creep model of *Culling*, 1963, or the viscous flow model of *Ahnert*, 1976) and, if the soil is explicitly modelled at all, diffusion is considered independent of depth through the profile. Simulation of the soil profile in mARM5D is novel as it explicitly calculates diffusive transport for each soil profile layer. Based on *Roering* (2004), the diffusivity is assumed to decrease exponentially with depth below the soil surface:

$$Dc_l = \exp(-\lambda h_l) \tag{3}$$

where $Dc_l$ is the fraction of diffusion rate for the layer $l$ relative to the diffusion rate at the surface layer ($l_s$), $h_l$ is the mean depth (m) of profile layer $l$ relative to the surface and $\lambda$ is a calibration parameter. We used $\lambda$=0.02 based on *Fleming and Johnson* (1975) and *Roering* (2004). The surface diffusion sediment transport rate ($D_s$; m) is:

$$D_s = \left(\frac{S}{S_a}\right)^{\beta} D_o \Delta t \tag{4}$$

where $D_o$ is the surface diffusivity (m/y) and $S_a$ is the adjustment slope, the average slope in which $D_o$ was measured/estimated. Here we use $S_a$=0.2 which approximate our field site average slope. Using an extensive sensitivity analysis we have found that $\beta$=0.1 yielded the best approximation to our field site's soil distribution. This value differs from the typical assumption of a linear relationship between slope and diffusion ($\beta$=1), suggesting that the influence of topographic slope in this soilscape is much lower. We will discuss this later. The removal of material due to diffusion from a given layer is calculated as the proportion of the layer's movable material (expressed in the model as a PSD vector $g_l$) that has been displaced at each iteration:

$$g_{l_{t+1}} = g_{l_t} - \left[ g_{l_t} \left(\frac{D_s}{\sqrt{A_p}}\right) Dc_l \right] \tag{5}$$

**2.1.3 Aeolian deposition**

Sediment, with a user-defined grading distribution ($g_a$), is added to the surface layer. The aeolian deposition rate ($K_a$; mm/yr) is assumed to be spatially uniform:

$$h_s g_{s_{t+1}} = h_s g_{s_t} + K_a g_a \tag{6}$$

where $g_{s_t}$ is the vector for the surface layer PSD and $h_s$ is the thickness of the surface layer. We use the same PSD as in the fine-grained simulation in Cohen et al., (2015), with a $d_{50}$=0.06 mm (derived from Bruins and Yaalon, 1992)

Sagy Cohen 11/21/2016 11:20 AM

Sagy Cohen 11/21/2016 11:20 AM

Sagy Cohen 11/21/2016 11:20 AM

Sagy Cohen 11/21/2016 11:20 AM

Sagy Cohen 11/21/2016 11:20 AM

Sagy Cohen 11/21/2016 11:20 AM

Sagy Cohen 11/21/2016 11:20 AM

Sagy Cohen 11/21/2016 11:20 AM

Sagy Cohen 11/21/2016 11:20 AM

Sagy Cohen 11/21/2016 11:20 AM

Sagy Cohen 11/21/2016 11:20 AM

Sagy Cohen 11/21/2016 11:20 AM

[revised manuscript text omitted]